# Architecture and RNA binding of the human negative elongation factor

Seychelle M Vos[1†], David Pöllmann[1,2,3†], Livia Caizzi[1], Katharina B Hofmann[1], Pascaline Rombaut[2,3], Tomasz Zimniak[2,3], Franz Herzog[2,3], Patrick Cramer[1*]

[1]Department of Molecular Biology, Max Planck Institute for Biophysical Chemistry, Göttingen, Germany; [2]Gene Center Munich, Ludwig-Maximilians-Universität München, Munich, Germany; [3]Department of Biochemistry, Ludwig-Maximilians-Universität München, Munich, Germany

**Abstract** Transcription regulation in metazoans often involves promoter-proximal pausing of RNA polymerase (Pol) II, which requires the 4-subunit negative elongation factor (NELF). Here we discern the functional architecture of human NELF through X-ray crystallography, protein crosslinking, biochemical assays, and RNA crosslinking in cells. We identify a NELF core subcomplex formed by conserved regions in subunits NELF-A and NELF-C, and resolve its crystal structure. The NELF-AC subcomplex binds single-stranded nucleic acids in vitro, and NELF-C associates with RNA in vivo. A positively charged face of NELF-AC is involved in RNA binding, whereas the opposite face of the NELF-AC subcomplex binds NELF-B. NELF-B is predicted to form a HEAT repeat fold, also binds RNA in vivo, and anchors the subunit NELF-E, which is confirmed to bind RNA in vivo. These results reveal the three-dimensional architecture and three RNA-binding faces of NELF.

*For correspondence: patrick. cramer@mpibpc.mpg.de

†These authors contributed equally to this work

Competing interests: The authors declare that no competing interests exist.

## Introduction

Transcription of eukaryotic protein-coding genes by RNA polymerase II (Pol II) is not only regulated during the initiation phase (*Hahn and Young, 2011*; *Sainsbury et al., 2015*) but also during elongation (*Jonkers and Lis, 2015*; *Li and Gilmour, 2011*; *Yamaguchi et al., 2013*). For many metazoan genes, elongating Pol II pauses near the promoter, about 20-60 base pairs downstream of the transcription start site (TSS) (*Kwak and Lis, 2013*). Such promoter-proximal pausing is a key event during post-initiation regulation of transcription (*Muse et al., 2007*; *Zeitlinger et al., 2007*). Genes involved in cellular responses, differentiation, and reprogramming are subject to regulation at the step of promoter-proximal pausing (*Williams et al., 2015*; *Min et al., 2011*).

Promoter-proximal pausing requires the DRB sensitivity-inducing factor (DSIF), a heterodimer of subunits Spt4 and Spt5 (*Wada et al., 1998*; *Yamaguchi et al., 1999b*). DSIF binds over the active site cleft of the Pol II elongation complex to encircle nucleic acids bound in the cleft (*Martinez-Rucobo et al., 2011*; *Klein et al., 2011*). Promoter-proximal pausing also employs the negative elongation factor (NELF) (*Pagano et al., 2014*; *Yamaguchi et al., 1999a*), which comprises the four subunits NELF-A, -B, -C (or its variant -D, which lacks nine N-terminal amino acid residues), and -E (*Narita et al., 2003*).

DSIF and NELF assemble early with transcribing Pol II, which leads to stable promoter-proximal pausing (*Henriques et al., 2013*; *Wu et al., 2003*; *Yamaguchi et al., 1999a*). Nucleosomes may also contribute to pausing (*Gilchrist et al., 2008*, *2010*; *Jimeno-González et al., 2015*). Pol II pause release relies on a kinase complex called positive transcription elongation factor b (P-TEFb) (*Chiba et al., 2010*). P-TEFb phosphorylates DSIF, NELF and the Pol II C-terminal domain (CTD), which encodes 52 heptapeptide repeats (consensus sequence $Y_1S_2P_3T_4S_5P_6S_7$) that are variably

phosphorylated during the transcription cycle (*Buratowski, 2009*; *Cheng and Price, 2007*; *Fujinaga et al., 2004*; *Yamada et al., 2006*).

DSIF, P-TEFb and NELF are differentially conserved among eukaryotes. DSIF is the most widely conserved complex with homologs present in most eukaryotes and even prokaryotes. P-TEFb homologs are found in most eukaryotes, whereas NELF conservation is more limited. For example, NELF homologs have not been identified in model organisms such as *Arabidopsis thaliana, Saccharomyces cerevisiae,* and *Caenorhabditis elegans* (*Narita et al., 2003*).

NELF was identified as a complex that cooperates with DSIF to repress Pol II elongation (*Yamaguchi et al., 1999a*). NELF apparently requires a preformed Pol II-DSIF elongation complex for stable binding (*Missra and Gilmour, 2010*; *Narita et al., 2003*; *Yamaguchi et al., 2002*). NELF binding efficiency and the rate of transcription elongation define the position of Pol II pausing (*Li et al., 2013*). NELF is associated with chromatin (*Wu et al., 2005*) and represses transcription elongation (*Wu et al., 2003*; *Yamaguchi et al., 2002*). The NELF-E subunit contains a RNA recognition motif (RRM) domain that binds RNA (*Pagano et al., 2014*; *Rao et al., 2006*; *2008*), and this may contribute to pausing (*Yamaguchi et al., 2002*; *Fujinaga et al., 2004*). NELF is required for the regulation of the heat stress gene *hsp70* (*Wu et al., 2003*), immediate early genes such as *junb* (*Aida et al., 2006*), and expression of genes of the human immunodeficiency virus (*Natarajan et al., 2013*; *Zhang et al., 2007*). It was recently suggested that NELF binds to enhancer RNAs (*Schaukowitch et al., 2014*).

Despite the central role of promoter-proximal pausing in gene regulation, the molecular mechanisms for Pol II pausing are unknown. Elucidating this mechanism requires structural information of the NELF complex. Here we show that regions of human NELF-A and NELF-C form a highly conserved core subcomplex with a novel fold. One side of this NELF-AC subcomplex exhibits a conserved binding face for single-stranded nucleic acids. RNA binding experiments in human cells reveal that NELF-B, NELF-C, and NELF-E associate with RNA in vivo. Our data provide the first structural model of NELF and extend our understanding of its RNA binding surfaces.

## Results

### NELF subcomplex NELF-AC

In a long-standing effort to obtain structural information on the intrinsically flexible NELF complex, we delineated regions in human NELF subunits that form soluble subcomplexes amenable to structural analysis (*Figure 1A*, *Table 1*, *Figure 1—figure supplement 1*, Materials and methods). Bacterial co-expression of NELF subunit variants revealed that the N-terminal region of NELF-A could be co-purified with NELF-C. Limited proteolysis and co-expression analysis with truncated protein variants showed that the N-terminal residues 6–188 of human NELF-A and residues 183–590 of human NELF-C formed a stable subcomplex ('NELF-AC') (*Figure 1—figure supplement 1B*). Purified NELF-AC could be crystallized by vapor diffusion, and the X-ray structure was solved by single isomorphous replacement with anomalous scattering (SIRAS) (*Figure 1—figure supplement 1C–E*, Materials and methods). The structure contained one NELF-AC heterodimer in the asymmetric unit and was refined to a free *R*-factor of 25.6% at 2.8Å resolution (*Table 2*). The structure shows very good stereochemistry and lacks only the mobile NELF-A residues 183–188, and NELF-C residues 183–185, 401–402, 445–448, 523, and 564–572.

### Unusual structure of the NELF-AC subcomplex

The structure of human NELF-AC reveals a novel fold and an extended interface between the two NELF subunits (*Figures 1B*, *2*). NELF-C adopts a horseshoe-like structure (*Figure 2A*). NELF-C consists of 22 α-helices (α1′-α22′) and a small two-stranded β-sheet (β1′-β2′, residues 367–379) that protrudes from the surface. The C-terminal half of NELF-C (helices α14′-α19′) forms three HEAT repeats (H1-H3). The HEAT repeat region shows structural similarity (*Holm and Rosenstrom, 2010*) to the C-terminal repeat domain (CTD)-interacting domain (CID) (*Meinhart and Cramer, 2004*) and the polyadenylation factor symplekin (*Xiang et al., 2010*). Despite the presence of a CID-like fold, NELF-AC did not show significant binding to CTD diheptad peptides carrying phosphorylations at CTD residues serine-2, serine-5, serine-2 and serine-5, or a consensus non-phosphorylated CTD diheptad peptide (not shown). Subunit NELF-A forms a highly conserved helical 'N-terminal domain' (helices α1–α5, residues 6–110)

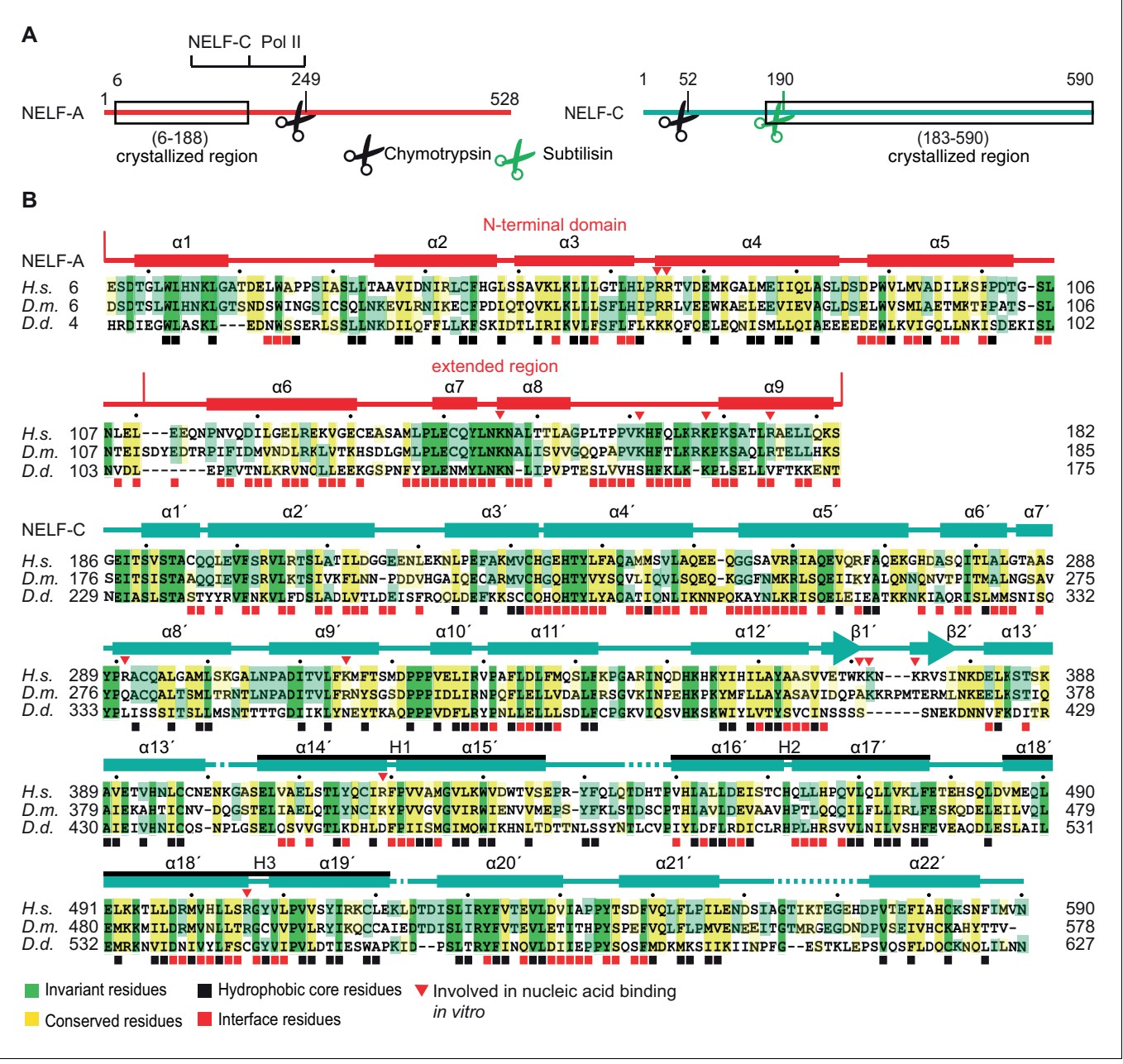

**Figure 1.** Primary structure and conservation of human NELF-A and NELF-C. (A) Crystallized variant and previously identified functional regions in human NELF-A and NELF-C. Cleavage sites of chymotrypsin and subtilisin are indicated by black and green scissors, respectively. 'NELF-C' delineates the previously identified NELF-C-binding region in NELF-A (**Narita et al., 2003**), whereas 'Pol II' marks the region in NELF-A that associates with Pol II (**Narita et al., 2003**). Boxed regions indicate crystallization constructs. (B) Alignment of NELF-A and NELF-C regions present in the structure from *Homo sapiens (H.s.), Drosophila melanogaster (D.m.)* and *Dictyostelium discoideum (D.d.)*. Invariant and conserved residues are highlighted in green and yellow, respectively. Lighter shades of green or yellow indicate conservation between only two of the represented organisms. Barrels above the alignment represent α-helices, arrows β-sheets. HEAT-repeats H1-H3 are marked with black lines above the alignment. Residues in the heterodimeric interface and hydrophobic core are marked by black and red squares, respectively. Red triangles label residues potentially involved in nucleic acid interaction as identified here. The 'N-terminal domain' and 'extended region' of NELF-A are indicated. Sequence alignments were carried out with ClustalW2 (**Larkin et al., 2007**) followed by manual editing and rendered with JALVIEW (**Waterhouse et al., 2009**).

The following figure supplements are available for figure 1:

**Figure supplement 1.** Iterative truncation of full-length NELF-AC yields a variant amenable to crystallization and region of the electron density map.

*Figure 1 continued on next page*

*Figure 1 continued*

**Figure supplement 2.** Multiple sequence alignment of full-length NELF-A demonstrating the comparatively high conservation of the crystallized region.

**Figure supplement 3.** Multiple sequence alignment of the N-terminal region of NELF-C.

that resembles (*Holm and Rosenstrom, 2010*) the fold of the HIV integrase-binding domain present in human PC4 and SFRS1-interacting protein (PSIP1) (*Cherepanov et al., 2005*) (*Figure 2B*). This domain is followed by an 'extended region' in NELF-A that forms four additional helices (helices α6-α9, residues 111–182) arrayed around the NELF-C horseshoe (*Figures 1B*, *2A*).

Both NELF-A regions interact extensively with NELF-C through hydrophobic and polar contacts. Two invariant tryptophan side chains (W24 and W89) on the NELF-A N-terminal domain insert into largely conserved hydrophobic pockets of NELF-C (*Figures 1B*, *2C*). The extended region of NELF-A is essential for NELF-C interaction (*Narita et al., 2003*) and contacts the N- and C-terminal regions of NELF-C with its helices α6 and α9, respectively. NELF-A helices α7 and α8 are buried in the NELF-C horseshoe (*Figure 2D*). Overall, the heterodimer interface has a large surface area (3690 $\text{Å}^2$), explaining the stability of the complex in 2 M sodium chloride (not shown).

## The NELF-AC core is highly conserved

The crystallized regions of human NELF-AC share considerable homology among metazoans, particularly at residues forming the hydrophobic cores and the interface between NELF-A and NELF-C (*Figure 1B*). The extent of conservation is evident when human and *Drosophila melanogaster* are compared, which share 55% identity for NELF-A and 50% identity for NELF-C. Intriguingly, NELF-A and -C homologs are present in some worms such as the filiarial nematode *Loa loa* (*Figure 1—figure supplements 2*, *3*) and single celled organisms such as the green algae *Chlorella variabilis* and the slime mold *Dictyostelium discoideum* (*Figure 1—figure supplements 2*, *3*). Most regions outside of the crystallized NELF-AC core diverge between single celled organisms and metazoans (*Figure 1—figure supplements 2*, *3*). Such conservation suggests that NELF may have been present in early eukaryotes and was lost in certain lineages over time.

## NELF-AC structure is preserved in the complete NELF complex

To place the NELF-AC crystal structure in context of the NELF tetramer, we determined the architecture of the four-subunit NELF complex by lysine specific crosslinking followed by mass spectrometry. We expressed the full-length four-subunit NELF complex recombinantly in insect cells from a single virus and purified it to homogeneity (Materials and methods, *Figure 3A*). The purified complex was

**Table 1.** Solubility of bacterially expressed NELF variants. Variants are full-length proteins if not otherwise specified. (+) = low solubility, (++) = medium solubility, (+++) = high solubility, (1) = aggregation, (2) = slight aggregation, (3) = stable at high salt concentrations only (500 mM NaCl).

| Protein variant | Solubility |
| --- | --- |
| NELF-AC | (+), (1) |
| NELF-AD | (++), (1) |
| NELF-AC$_{36-590}$ | (++), (1) |
| NELF-A$_{6-188}$C$_{36-590}$ | (+++), (2) |
| NELF-A$_{6-188}$C$_{183-590}$ | (+++) |
| NELF-ABC | (++), (1) |
| NELF-A$_{6-188}$BC$_{36-590}$ | (++), (3) |
| NELF-ABCE | (+), (1), (3) |
| NELF-A$_{6-188}$BC$_{36-590}$E | (++), (2), (3) |

**Table 2.** X-ray diffraction and refinement statistics.

| | Native | SeMet |
|---|---|---|
| **Data collection[a]** | | |
| Space group | I213 | I213 |
| Unit cell dimensions | | |
| $a=b=c$ (Å) | 185.07 | 184.45 |
| Unit cell angles | | |
| $\alpha=\beta=\gamma$ (°) | 90 | 90 |
| Wavelength (Å) | 1.00000 | 0.97910 |
| Resolution (Å) | 46–2.75 (2.82–2.75)[b] | 43–3.25 (3.33–3.25) |
| $R_{sym}$ (%) | 9 (271) | 9 (130) |
| $I/\sigma$ | 32.2 (1.9) | 24.0 (2.2) |
| Completeness (%) | 100 (100) | 100 (100) |
| Redundancy | 39.8 (40.7) | 20.6 (15.8) |
| $CC_{(1/2)}$ (%) | 100 (73.9) | 100 (78.4) |
| No. reflections observed | 1,093,935 | 657,156 |
| No. Reflections unique | 27,492 | 31,923 |
| Figure of merit (SeMet sites) | | 0.323 |
| | | |
| **Refinement** | | |
| Resolution (Å) | 46-2.75 | |
| $R_{work}/R_{free}$ (%) | 23.7 (40.6) / 25.6 (43.3) | |
| No. atoms | | |
| Protein | 4434 | |
| Ligand/ion | 2 | |
| Water | 13 | |
| $B$ factors (Å$^2$) | | |
| Protein | 110.8 (NELF-A) 108.4 (NELF-C) | |
| Ligand/ion | 103.5 | |
| Water | 77.4 | |
| R.m.s.d. | | |
| Bond lengths (Å) | 0.003 | |
| Bond angles (°) | 0.662 | |
| Ramachandran Plot[d] | | |
| Allowed (%) | 1.81 | |
| Favored (%) | 98.19 | |
| Outliers (%) | 0 | |

[a]Diffraction data were collected at beamline X06DA of the Swiss Light Source, Switzerland and processed with XDS (**Kabsch, 2010**).[b]Values in parentheses are for the highest-resolution shells.[c]$CC_{1/2}$ = percentage of correlation between intensities from random half-datasets (**Karplus and Diederichs, 2012**).[d]Ramachandran plot categories were defined by Molprobity (**Chen et al., 2010**).

crosslinked with disuccinimidyl suberate (DSS) and lysine-lysine crosslinks were detected by mass spectrometry as previously described (**Herzog et al., 2012**). We obtained a total of 424 unique high-confidence lysine-lysine crosslinks, including 279 intersubunit and 145 intrasubunit crosslinks (**Figure 3—figure supplement 1A,B**, **Figure 3—source data 1**).

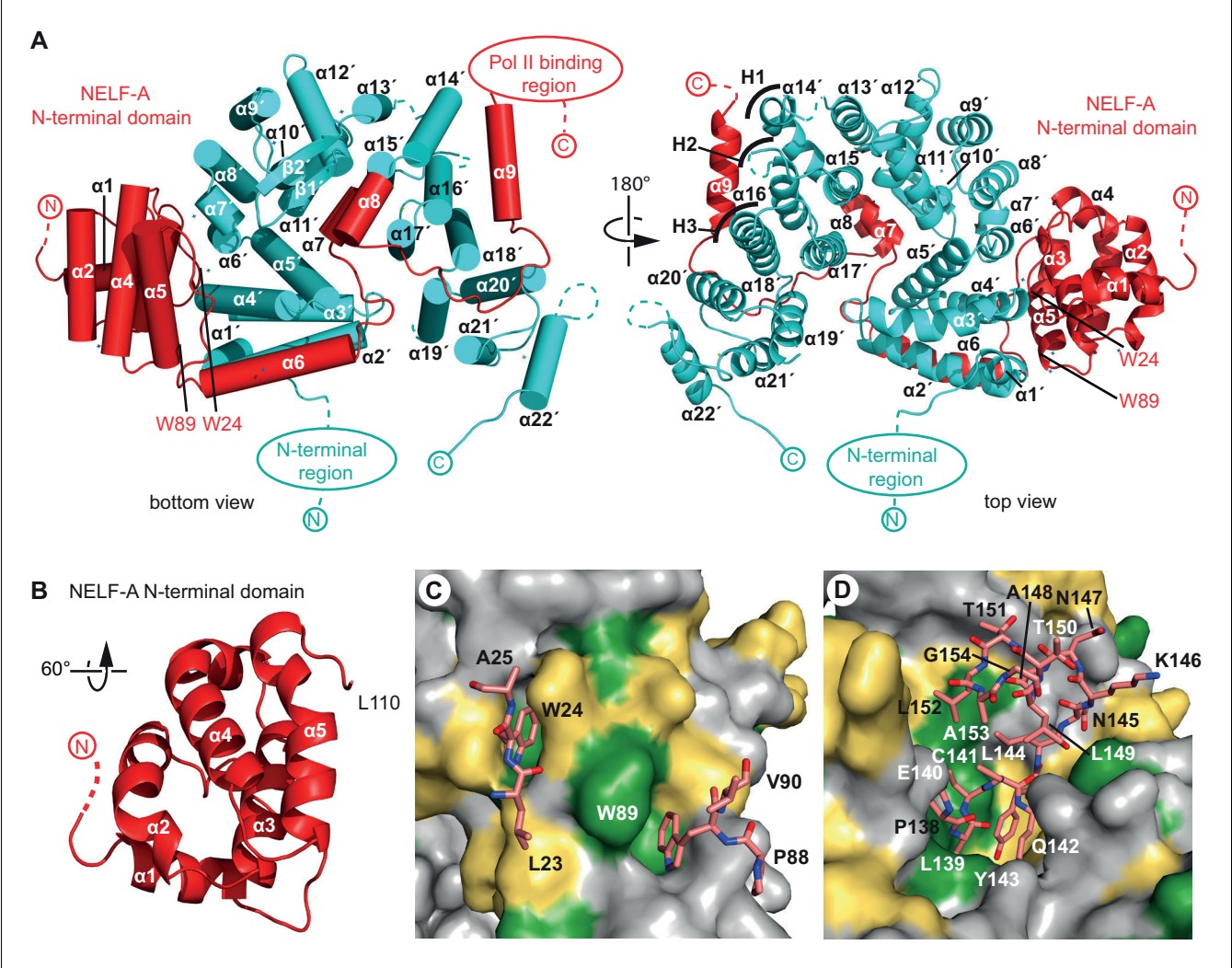

**Figure 2.** Crystal structure of human NELF-AC complex. (**A**) Ribbon model of NELF-AC with NELF-A in red and NELF-C in cyan. N- and C-termini, mobile regions, and truncated regions are indicated by dashed lines. The two views are related by a 180° rotation around the vertical axis. Curved lines marked 'H1–H3' demarcate alpha helices involved in heat repeats 1–3 (1: 14', 15'; 2: 16', 17'; 3: 18', 19'). Alpha helices are named as in *Figure 1B*. All crystallography figures were rendered with Pymol (*PyMOL, 2002*). (**B**) NELF-A N-terminal domain enlarged and rotated 60° around the horizontal axis relative to 'bottom view' (*Figure 2A*). (**C**) Detailed view of invariant NELF-A residues W24 and W89 and surrounding residues (stick model) interacting with the NELF-C surface. NELF-C surface conservation colored according to *Figure 1B*. The view is rotated by 90° around the vertical axis relative to 'bottom view' (*Figure 2A*). (**D**) Detailed view of NELF-A helices α7 and α8 (stick model, residues 138–154) surrounded by NELF-C. NELF-C surface conservation is colored according to *Figure 1B*. The view is rotated 60° around the horizontal axis relative to 'bottom view' (*Figure 2A*).

Our NELF-AC crystal structure explained 11 inter- and intrasubunit crosslinks, with Cα distances below the maximum allowed distance of 30 Å (*Figure 3C*, *Figure 3—figure supplement 1C*). We detected only one NELF-A and five NELF-C intrasubunit crosslinks that exceeded a Cα distance of 30 Å, and these could generally be explained by local flexibility. However, for NELF-C, the >30 Å intrasubunit crosslinks occur between α helices 12' and 13' (K353, K380, K384, K388) and α helices 18' and 19' (K494, K518) (*Figures 2, 4A*). These crosslinks suggest helices 12' and 13' may change conformation. Together these data indicate that the structure of NELF-AC is largely preserved within the complete NELF complex.

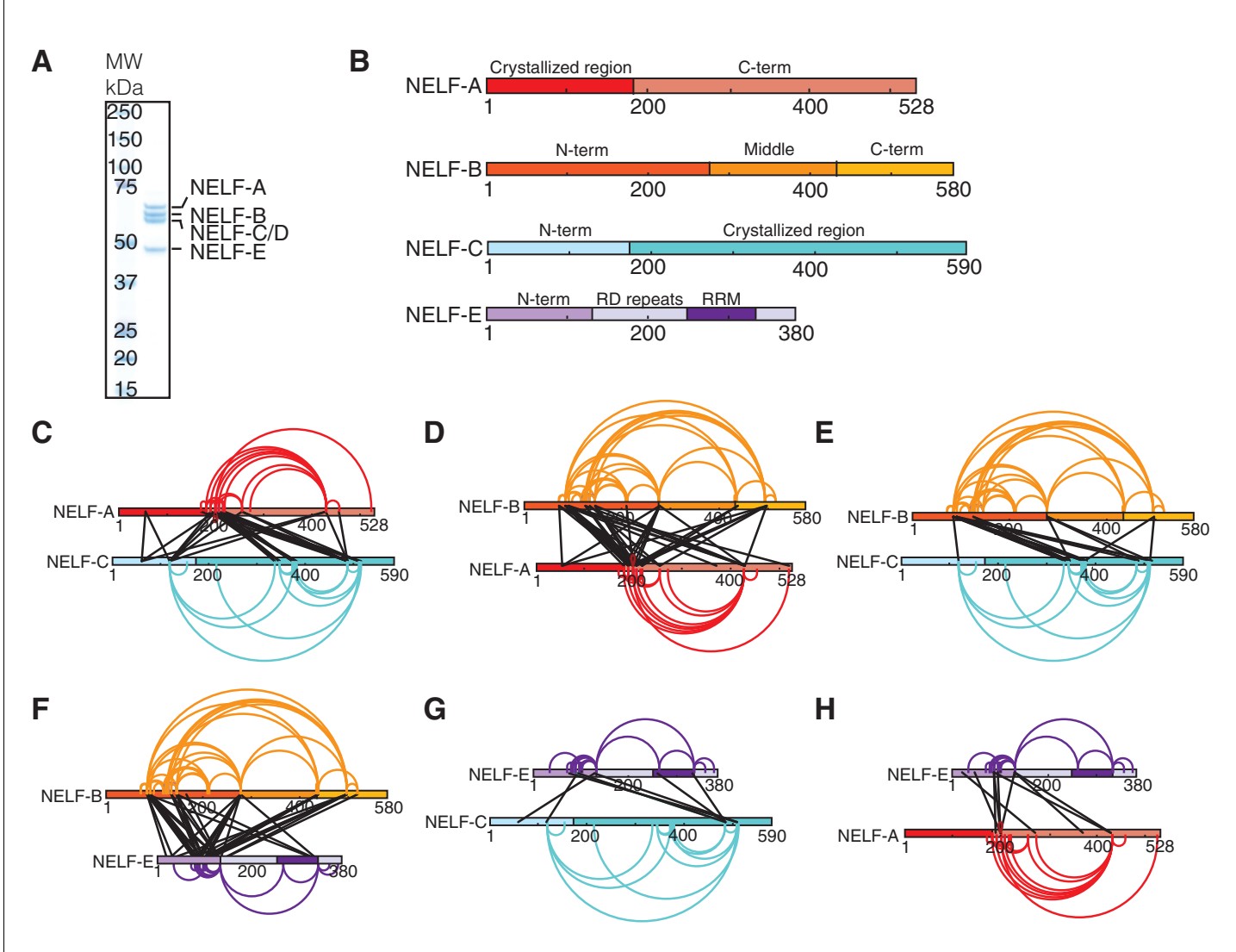

**Figure 3.** Architecture of human NELF complex as detected by crosslinking MS. (**A**) The four-subunit human NELF complex was recombinantly expressed in insect cells and purified to homogeneity. The purified complex (0.9 μg) was run on a 4–12% gradient sodium dodecyl sulfate polyacrylamide gel electrophoresis (SDS-PAGE) and stained with Coomassie blue. A molecular weight marker (MW) in kilodaltons (kDa) is provided on the left side of the gel. (**B**) Cartoon depiction of individual NELF proteins. Different shades designate unique regions or domains of each protein (NELF-A red, NELF-B orange, NELF-C cyan, NELF-E purple). N- and C-term refers to the N- or C-terminus of a protein, respectively. RD repeats refer to a flexible region of NELF-E that is primarily composed of Arg and Asp residues. (**C–H**) Crosslinks detected within the NELF tetramer displayed as binary interactions. Intraprotein crosslinks are shown as curves and colored as in (**B**). Interprotein crosslinks are shown as black lines. The endpoint of each line specifies a specific residue in the corresponding protein. Crosslinking data is filtered to display crosslinks with an ID-score greater than 30. A full map of all interprotein crosslinks is provided in *Figure 3—figure supplement 1A*. All crosslinks can be found in *Figure 3—source data 1*. Crosslinking data modeled with xiNET (*Combe et al., 2015*). (**C**) NELF-A, NELF-C (**D**) NELF-B, NELF-A (**E**) NELF-B, NELF-C (**F**) NELF-B, NELF-E (**G**) NELF-E, NELF-C (**H**) NELF-E, NELF-A.

The following source data and figure supplement are available for figure 3:

**Source data 1.** Crosslinking MS of four-subunit NELF complex primary data for intra and interprotein crosslinks.

**Figure supplement 1.** Additional information supporting NELF architecture as determined by crosslinking MS.

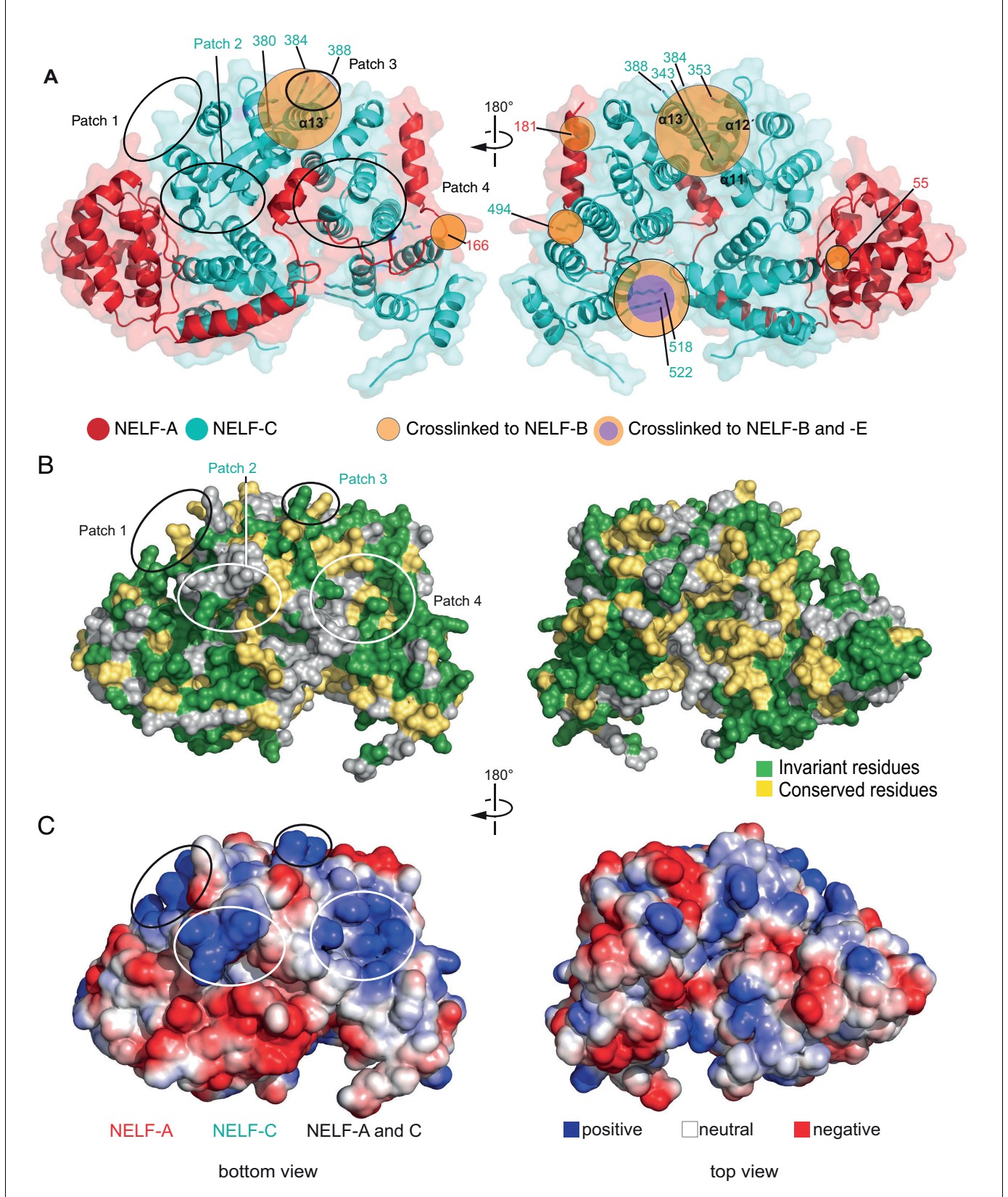

**Figure 4.** Surface properties of NELF-AC. Two views of the solvent-accessible surface related by 180° rotations around a vertical axis are shown. (**A**) Surface representation of NELF-AC with shaded ellipses representing regions where NELF-B (orange) or NELF-E (purple) crosslinks with NELF-AC were

*Figure 4 continued on next page*

*Figure 4 continued*

detected. Lysines are shown in stick representation and numbered. NELF-B and NELF-E primarily crosslink to one face of the NELF-AC dimer. Four empty ellipses represent patches of positively charged residues. (**B**) Surface conservation. Residues that are invariant from human to *Drosophila* are in green, conserved residues in yellow (***Figure 1B***). Surface areas involved in nucleic acid binding (patches 1–4) (Results) are highlighted. Colors of labels according to color code of protein features belong to (***Figure 2***). (**C**) Electrostatic surface potential generated with ABPS (***Baker et al., 2001***). Blue, red, and white areas indicate positive, negative and neutral charge, respectively. Surface areas involved in nucleic acid binding (patches 1–4). Colors of labels according to color code of protein features belong to (***Figure 2***).

The following figure supplement is available for figure 4:

**Figure supplement 1.** Conserved surface patch 5.

## Topology of the NELF complex

We next used our crosslinking data to determine the topology of the complete NELF complex. NELF-B is the only subunit of the NELF complex with no available structural information. We first used our crosslinking data to determine the relative architecture of NELF-B. Intrasubunit crosslinks reveal three distinct modules that we identify as the N-terminal, middle, and C-terminal regions. The N-terminal region of NELF-B (85–291) forms extensive crosslinks within itself and with the C-terminal region (438–519). The C-terminal region also crosslinks considerably with itself. The middle region (291–438) forms intrasubunit crosslinks within but not outside of the module, suggesting that it may act as a hinge between the N- and C- terminal regions.

We next used the program I-TASSER to generate homology based models of NELF-B (***Kelley et al., 2015***; ***Yang et al., 2015***). I-TASSER predicted that NELF-B forms a HEAT repeat fold (C-score = -2.31, best template structure is 1B3U – human PP2A). The model is supported by our crosslinking data, suggesting a strong curvature of the HEAT repeat fold, as observed for a HEAT repeat protein folding around its interaction partner (***Cingolani et al., 1999***) (***Figure 3—figure supplement 1D***).

NELF-B crosslinks with NELF-AC via its N- and C- terminal regions. The crosslinks primarily map to a single face of NELF-AC (***Figure 4A***). Two NELF-A residues present in the NELF-AC crystal structure, K55 and K166 form crosslinks with NELF-B residues K72, K278, K487 and K85 (***Figure 3D***). NELF-C primarily crosslinks to the N-terminal region of NELF-B (K85, K92, K126, K146) via $\alpha$ helices 12′ and 13′ and helices 18′ and 19′ (***Figure 3E***). Interestingly, one N-terminal residue of NELF-C that is not present in our crystal structure (K125) forms three crosslinks with NELF-B (K85, K278, and K497). NELF-B and -E crosslink extensively, consistent with biochemical interaction data (***Narita et al., 2003***) (***Figure 3F***). The N-terminus and the RRM domain of NELF-E (***Rao et al., 2006***) (residues 257–335, PDB ID: 2JX2) crosslink to both the N- and C-terminal regions of NELF-B. We also detect an intrasubunit crosslink between NELF-E residues K260 and K332, which are located on the same face of the RRM (***Figure 3—figure supplement 1E***), supporting the general conservation of the fold in the complex.

With respect to crosslinks between NELF-E and NELF-C, two lysines in the non-crystallized N-terminal region of NELF-C (K66 and K125) form several crosslinks with NELF-E, including its RRM domain. Regions of NELF-C in the vicinity of helices 18′ and 19′, which are also responsible for some of the detected crosslinks between NELF-B and -C, crosslink with the NELF-E N-terminal region and the RRM (***Figures 3G***, ***4A***). No crosslinks were detected between the crystallized region of NELF-A and NELF-E (***Figure 3H***). The region directly following the crystallized portion of NELF-A (190–255) forms multiple crosslinks with the rest of the NELF complex. Given that NELF-A (190–255) is highly susceptible to proteolysis and is predicted to be the primary region associating with Pol II, the multiple interprotein crosslinks formed by NELF-A (190–255) with NELF-B, -C, and -E are likely favored by flexibility within the molecule when Pol II is absent (***Narita et al., 2003***).

Our crosslinking data suggest the 3D topology for the entire NELF complex. The NELF-AC subcomplex interacts with NELF-B primarily through contacts made by NELF-C and the N-terminal region of NELF-B. The opposite face of NELF-AC remains solvent exposed in the complex. NELF-B with its predicted heat repeats forms a cradle around the N-terminal region of NELF-E, tethering the RRM domain to the rest of the complex. Taken together, NELF is a modular, flexible, multivalent complex with many interaction faces for both nucleic acids and protein partners.

## NELF-AC binds single-stranded nucleic acids

Analysis of the surface of the NELF-AC structure showed that the face opposite of where we detected NELF-BE crosslinks contains four positively charged patches (*Figure 4A–C*, bottom view). Patch 1 consists of NELF-A residues R65 and R66 and NELF-C residues R291 and K315. Patch 2 encompasses NELF-C residues K371, K372, and K374, and patch 3 contains NELF-C residues K384 and K388. Patch 4 is composed of NELF-A residues K146, K161, K168, and R175, and NELF-C residues R419 and R506. These patches are well conserved among metazoans, and are partially conserved in *Dictyostelium* (*Figure 1B*). Our crosslinking MS data revealed that all positive patches, except for patch 3 (NELF-C K384, K388), were devoid of crosslinks with NELF-B or -E indicating that surface patches 1, 2, and 4 are not involved in subunit contacts. In addition to the four positive patches, the NELF-AC surface contains a conserved polar patch (patch 5) that is formed by NELF-A residues K166, R167, K170, L174, E177, K181, and S182, and residues E491, K494, D498, D526, S528, R531, Y532, T535 and E536 that protrude from NELF-C helices α18' and α20' (*Figure 4—figure supplement 1*). Interestingly, three lysines in this region (NELF-A K166, K181, NELF-C K494) crosslink to NELF-B.

The positively charged patches of NELF-AC suggested that the subcomplex may associate with nucleic acid. To investigate this, we used fluorescence anisotropy titration assays (*Figure 5*, Materials and methods). We first assessed NELF-AC binding to 25-nt, single-stranded (ss) DNA and ssRNA oligonucleotides bearing a 5' FAM label. Two random sequences with either 44% or 60% GC content were employed. Interestingly, we detected moderate binding of NELF-AC to the ssDNA and ssRNA with 60% GC content. Fitting the resulting binding curves by linear regression analysis gave apparent $K_d$'s in the low micromolar range (*Figure 5A*, *Table 3*). The addition of competitor tRNA did not affect NELF-AC association with the 60% GC ssDNA/ssRNA indicating that the interaction is specific (*Figure 5—figure supplement 1A*). In contrast, we found that the 44% GC content RNA failed to associate significantly with the NELF-AC complex (*Figure 5B*). Additionally, NELF-AC did not associate with nucleic acid duplexes composed of the 60% GC sequence (DNA or DNA-RNA hybrids, not shown), suggesting that RNA and DNA binding by the subcomplex may be sequence and structure dependent.

To investigate whether the positively charged patches were involved in nucleic acid binding, we generated NELF-AC variants in which lysine and arginine residues in the patches were substituted with methionine and glutamine, respectively. Indeed, single-stranded nucleic acid binding to the 60% GC RNA and DNA constructs was impaired in variants with mutations in three or four of the positively charged patches (*Figure 5C*, *Table 3*). The strongest RNA binding defects are associated with mutations to patches 1 and 4, whereas mutations to patch 2 appear to have a greater impact on ssDNA association.

We also tested whether single-stranded nucleic acids corresponding to known Pol II in vivo pause sites could associate with NELF-AC (*Figure 5—figure supplement 1C* inset). A ssRNA oligonucleotide with a sequence corresponding to a promoter-proximal transcript from the *junB* gene bound NELF-AC with a $K_d$ of ~8.0 ± 0.9 μM, whereas ssDNA corresponding to the non-template strand in this region bound more weakly (*Aida et al., 2006*) (*Figure 5—figure supplement 1C*). Furthermore, ssRNA and ssDNA derived from the *c-fos* promoter-proximal region sequences (*Fivaz et al., 2000*) also bound NELF-AC, albeit with a preference for DNA (*Figure 5—figure supplement 1C*). Taken together, NELF-AC binds single-stranded nucleic acids in vitro via positively charged patches, and suggests both the strength of binding and the preference for RNA or DNA is possibly sequence-dependent.

## NELF-AC and NELF-B associate with RNA in context of the NELF tetramer

We next addressed whether NELF-AC associates with nucleic acids while residing in the NELF tetramer. The NELF-E RRM is reported to bind RNA in the mid nanomolar to micromolar range (*Pagano et al., 2014*; *Rao et al., 2006*) and thus could mask nucleic acid interactions by other subunits in our binding assays. To aid data interpretation, we generated NELF variants that lack the NELF-E RRM or NELF-E entirely. We used our crosslinking and limited proteolysis experiments to generate a NELF-E N-terminal fragment that stably binds NELF-B, but lacks the RRM (NELF-E residues 1–138). The WT NELF tetramer, NELF ΔRRM, and NELF-ABC were overexpressed in insect cells and purified to homogeneity (Materials and methods, *Figure 6A*, *Figure 6—figure supplement 1A*).

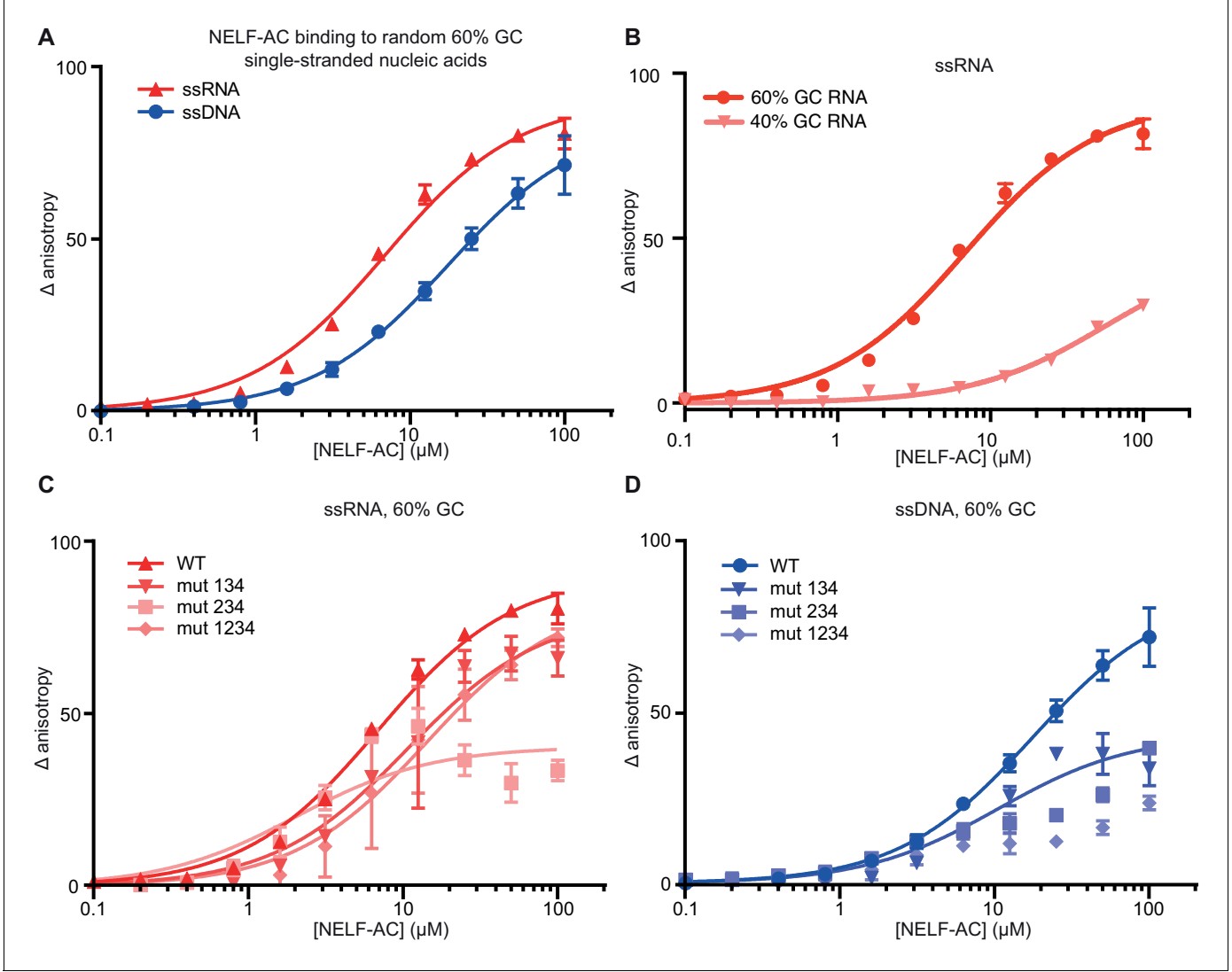

**Figure 5.** NELF-AC binds single-stranded nucleic acids. (**A**) Binding of wild type (WT) NELF-AC to 10 nM fluorescently labeled ssRNA or ssDNA with 60% GC content as monitored by the change in relative fluorescence anisotropy. Error bars reflect the standard deviation from three experimental replicates. (**B**) Binding of wild type (WT) NELF-AC to 10 nM fluorescently labeled ssRNA with 44% GC content. The 60% RNA binding data shown in (**A**) is shown as a reference. Error bars reflect the standard deviation from three experimental replicates. (**C**, **D**) Binding of WT NELF-AC and variants containing mutations in surface patches (*Figure 3*) to the same ssRNA (**B**) or ssDNA (**C**) used in panel A. Numbers indicate mutated patches present in NELF-AC variants. Patch 1: NELF-A R65Q, R66Q, NELF-C R291Q, K315M Patch 2: NELF-C K371M, K372M, K374M Patch 3: NELF-C K384M, K388M Patch 4: NELF-A K146M, K161M, K168M, R175Q NELF-C R419Q, R506Q.

The following figure supplement is available for figure 5:

**Figure supplement 1.** Fluorescence anisotropy controls.

The WT tetramer, NELF ΔRRM and NELF-ABC were subjected to fluorescence anisotropy titration assays using the labeled 25-nt 60% GC random RNA/DNA oligonucleotides employed for the NELF-AC studies. The WT protein binds both the 60% GC ssDNA and RNA, however, the resulting curves are complex and cannot be fit by a simple single site binding model (*Figure 6B*, *Figure 6—figure supplement 1B*). NELF ΔRRM and NELF-ABC also bind the 60% GC RNA, but the curves can be fit by a single site binding model (apparent $K_d$ NELF ΔRRM 30 ± 7 nM, NELF-ABC 75 ± 14 nM), demonstrating that regions of NELF outside of the NELF-E RRM associate with RNA. To further investigate NELF's RRM domain-independent RNA-binding behavior, a patch mutated variant of NELF-C

**Table 3.** Curve fitting data for NELF RNA and DNA binding as determined by fluorescence anisotropy. Fluorescence anisotropy data found in **Figures 5–7** were fit with a single site binding model where possible (Materials and methods). Apparent disassociation constants ($K_{d,app}$), $R^2$, and Bmax (maximum anisotropy) values with error where applicable for the 60% ssRNA and DNA substrate as well as the TAR RNA are shown. NA means fitting was not applicable.

**RNA 60% GC**

| Protein construct | $K_{d,app}$ (μM) | $R^2$ | Bmax |
|---|---|---|---|
| NELF-A (6-188)+ NELF-C (183–590) | 6.87 ± 0.46 | 0.99 | 91.66 ± 1.73 |
| Patch mutant 124 | 14.33 ± 2.41 | 0.91 | 79.26 ± 4.44 |
| Patch mutant 234 | NA | NA | NA |
| Patch mutant 1234 | 14.31 ± 2.67 | 0.93 | 84.94 ± 5.27 |
| WT NELF | NA | NA | NA |
| NELF ΔRRM | 0.030 ± 0.007 | 0.92 | 79.87 ± 3.67 |
| NELF-ABC | 0.074 ± 0.014 | 0.95 | 78.68 ± 3.66 |
| NELF+patch mutated -C | NA | NA | NA |
| NELF ΔRRM patch mutated -C | 0.094 ± 0.020 | 0.93 | 71.75 ± 3.99 |
| NELF-ABC patch mutated -C | 0.290 ± 0.99 | 0.85 | 76.99 ± 8.2 |
| NELF-B | 8.50 ± 1.59 | 0.96 | 113.1 ± 9.10 |
| NELF-B, NELF-E (1-138) | 2.83 ± 1.00 | 0.97 | 111 ± 4.42 |

**ssDNA 60% GC**

| Protein construct | $K_{d,app}$ (μM) | $R^2$ | Bmax |
|---|---|---|---|
| NELF-A (6-188)+ NELF-C (183-590) | 17.5 ± 1.33 | 0.98 | 85.66 ± 2.28 |
| Patch mutant 124 | 12.92 ± 2.28 | 0.93 | 12.92 ± 2.56 |
| Patch mutant 234 | NA | NA | NA |
| Patch mutant 1234 | NA | NA | NA |
| WT NELF | NA | NA | NA |
| NELF ΔRRM | 0.03 ± 0.01 | 0.71 | 39.05 ± 3.28 |
| NELF-ABC | 0.10 ± 0.03 | 0.85 | 43.68 ± 3.23 |
| NELF+patch mutated -C | NA | NA | NA |
| NELF ΔRRM patch mutated -C | 0.30 ± 0.08 | 0.92 | 50.26 ± 3.76 |
| NELF-ABC patch mutated -C | 0.77 ± 0.02 | 0.91 | 54.85 ± 7.60 |
| NELF-B | 6.31 ± 1.92 | 0.89 | 110.6 ± 13.15 |
| NELF-B, NELF-E (1-138) | 17.42 ± 3.5 | 0.97 | 170 ± 18.69 |

**TAR RNA Stem loop**

| Protein construct | $K_{d,app}$ (μM) | $R^2$ | Bmax |
|---|---|---|---|
| NELF-A (6-188)+ NELF-C (183-590) | NA | NA | NA |
| WT NELF | 0.146 ± 0.03 | 0.92 | 118.4 ± 4.50 |
| NELF ΔRRM | 0.869 ± 0.14 | 0.95 | 116.5 ± 4.72 |
| NELF-ABC | 1.32 ± 0.18 | 0.97 | 131.7 ± 5.22 |
| NELF-B, NELF-E (1-138) | 5.59 ± 1.05 | 0.95 | 82.82 ± 5.88 |

(residues R291Q, K315M, K371M, K372M, K374M, K384M, K388M, R419Q, R506Q) was used to replace the WT NELF-C protein in the WT, ΔRRM, and ABC complexes (**Figure 6—figure supplement 1C**). All NELF variants containing patch-mutated NELF-C showed reduced binding to RNA (**Figure 6C–E**). Binding deficits to the 60% GC RNA and ssDNA were similar in magnitude to those observed with the NELF-C patch mutations present in the NELF-AC subcomplex (**Figure 5**, **Figure 6—figure supplement 1D–F**, **Table 3**).

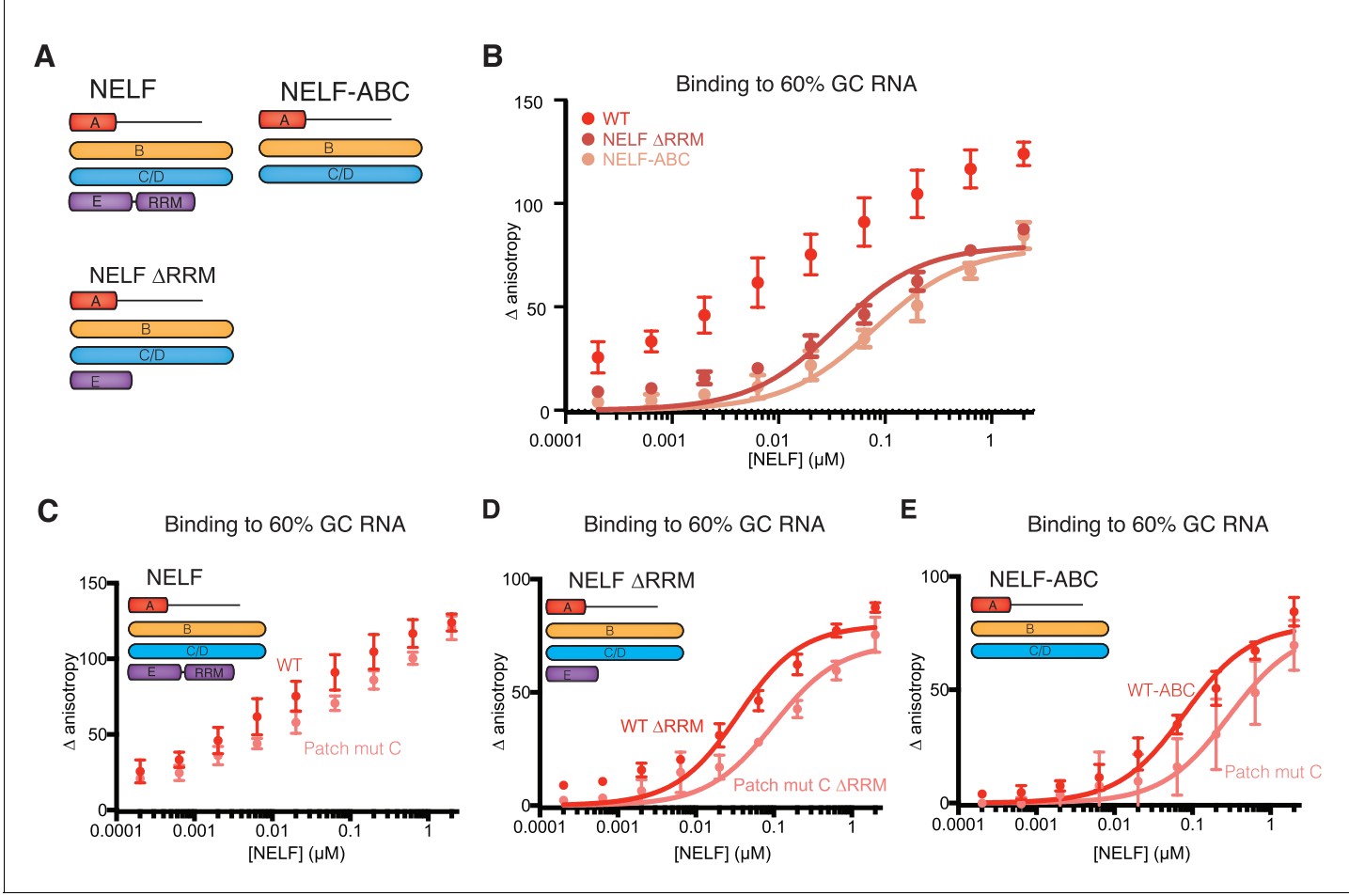

**Figure 6.** NELF-AC associates with RNA in context of the NELF tetramer. (**A**) NELF constructs expressed and purified from insect cells represented as cartoons. NELF was produced as a FL construct, a construct lacking the NELF-E RRM (NELF ΔRRM), or without NELF-E (NELF-ABC). (**B**) Binding of the WT NELF tetramer, NELF ΔRRM, or NELF-ABC to 10 nM of the 60% GC content RNA as determined by relative change in fluorescence anisotropy. Data was fit with a single site binding equation when possible. Apparent Kd values are found in *Table 3*. Error bars reflect the standard deviation from three experimental replicates. (**C–E**) Binding of the WT or patch mutated NELF-C variants to 10 nM of the 60% GC content RNA as determined by fluorescence anisotropy. The following residues were mutated in the NELF-C patch mutant: R291Q, K315M, K371M, K372M, K374M, K384M, K388M, R419Q, R506Q. Darker shades of red indicate the WT protein constructs whereas lighter shades of red indicate the NELF-C patch mutated variant. Error bars reflect the standard deviation from three experimental replicates. When possible, curves were fit with a single site binding model and apparent Kd values are found in *Table 3*. (**C**) NELF tetramer (**D**) NELF ΔRRM (**E**) NELF-ABC.

The following figure supplement is available for figure 6:

**Figure supplement 1.** Purity of NELF truncation constructs and DNA binding.

Despite containing the NELF-C patch-mutated variant, NELF ΔRRM and NELF-ABC retain the ability to bind RNA, suggesting that other regions of NELF-AC or NELF-B associate with RNA. To address whether NELF-B can associate with ss nucleic acid, NELF-B or NELF-B with an N-terminal fragment of NELF-E (1–138) were overexpressed in insect cells and purified (*Figure 6—figure supplement 1A*). Binding experiments performed with these NELF-B variants demonstrate that both bind the 60% GC RNA and ssDNA with affinities similar to that measured for NELF-AC (*Table 3*, *Figure 7A*). Together these data indicate that in addition to the NELF-E RRM, the NELF tetramer associates with RNA via NELF-B and -C.

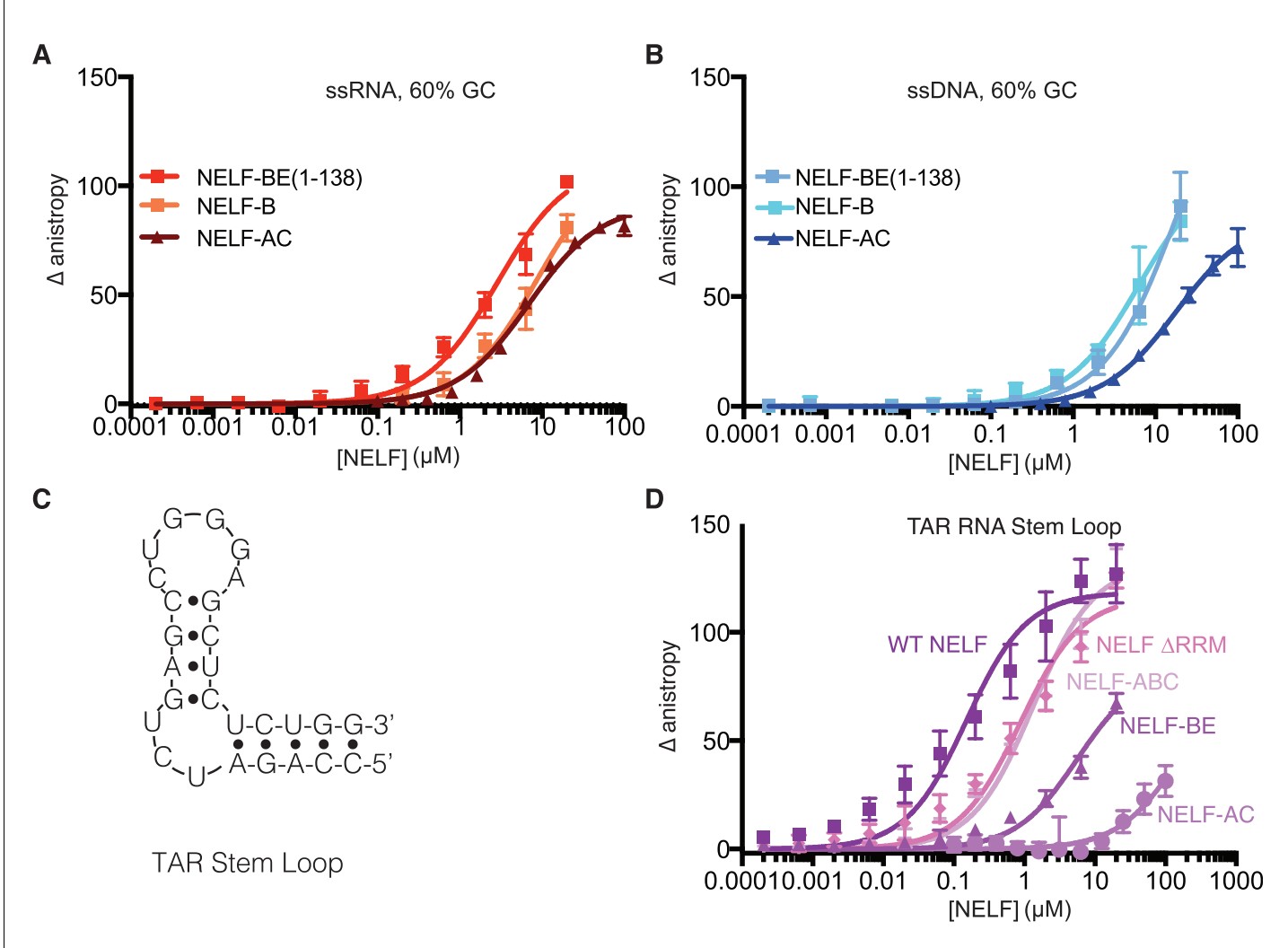

**Figure 7.** NELF-B association with ssRNA, ssDNA and TAR RNA stem loop. (**A**) Binding of NELF-B (light red) or NELF-BE (1–138) (red) to the 60% GC RNA as determined by fluorescence anisotropy. NELF-AC (dark red) (*Figure 5A*) binding to the same RNA is shown as a reference. Error bars reflect the standard deviation from three experimental replicates. Data were fit with a single site binding model. Apparent Kd values are reported in *Table 3*. (**B**) Binding of NELF-B (cyan) or NELF-BE (1–138) (sky blue) to the 60% GC DNA as determined by fluorescence anisotropy. NELF-AC (dark blue) (*Figure 5A*) binding is shown as a reference. Error bars reflect the standard deviation from three experimental replicates. Data were fit with a single site binding model. Apparent Kd values are reported in *Table 3*. (**C**) 2D structure of TAR RNA stem loop region used for fluorescence anisotropy experiments presented in (**D**). Dots indicate hydrogen bonds between bases. Lines represent the phosphate backbone. RNA was labeled with a 5' FAM label. (**D**) Binding of the NELF tetramer (dark purple), NELF ΔRRM (orchid), NELF-ABC (thistle), NELF-BE (1–138) (medium purple), and NELF-AC (light purple) to the TAR RNA stem loop. Data were fit with a single site binding model. Apparent Kd values are reported in *Table 3*.

## NELF-E and -B associate with the HIV-1 TAR stem loop

The human immunodeficiency virus (HIV)-1 transactivation response (TAR) element is a hairpin shaped RNA produced +59 nucleotides after Pol II initiates transcription from the HIV-1 long terminal repeat of the integrated HIV-1 virus (*Karn and Stoltzfus, 2012*) (*Figure 7C*). The TAR RNA is used to recruit P-TEFb and other factors to promoter proximally paused Pol II (*Ott et al., 2011*). The NELF-E RRM domain binds the HIV-1 TAR RNA in vitro (*Pagano et al., 2014*; *Rao et al., 2006*; *Yamaguchi et al., 2002*; *Fujinaga et al., 2004*) and is postulated to regulate Pol II elongation by associating with the TAR element (*Karn and Stoltzfus, 2012*). To expand our understanding of NELF association with the TAR RNA stem loop, we performed fluorescence anisotropy binding experiments with our collection of NELF variants and a 5' FAM labeled TAR stem loop. Binding

experiments performed with the WT NELF complex revealed strong association between NELF and the TAR RNA stem loop (146 ± 30 nM), similar to affinities reported by the Lis group for the isolated human NELF-E RRM and the TAR stem loop (200 ± 10 nM) (*Pagano et al., 2014*). Interestingly, NELF complexes lacking the NELF-E RRM or NELF-E retained the ability to associate with the TAR stem loop albeit with a ≈6–10-fold reduction in binding affinity (869 ± 140 nM NELF ΔRRM, NELF-ABC 1.32 ± 0.18 μM). To determine which subcomplex of NELF is responsible for the non-RRM mediated association with the TAR stem loop, we tested binding of NELF-AC and NELF-BE (1–138) to the TAR stem loop. NELF-BE (1–138) modestly associated with the TAR stem loop (5.6 ± 1.0 μM) whereas NELF-AC showed little association (*Figure 7D*). This further suggests that RNA binding by NELF-AC is influenced by RNA sequence and/or structure and that NELF associates with RNA on surfaces outside of the RRM domain.

### NELF-B, NELF-C and -E associate with RNA in vivo

It is known that the NELF-E subunit binds RNA in vitro and in vivo (*Pagano et al., 2014*; *Yamaguchi et al., 2002*; *Schaukowitch et al., 2014*; *Missra and Gilmour, 2010*). Our biochemical experiments demonstrate that NELF additionally binds to single-stranded nucleic acids via NELF-B and NELF-AC in vitro. To determine whether NELF-B or NELF-AC can also associate with RNA in vivo, we performed photoactivatable-ribonucleoside-enhanced crosslinking and immunoprecipitation experiments in Jurkat and 293FT cells. Cells were treated for 16 hr with 4-thiouridine (4sU) to label RNA and enhance crosslinking efficiency. RNA was crosslinked to associated proteins using UV light at a wavelength of 365 nm prior to immunoprecipitation with subunit-specific antibodies. The immunoprecipitated material was treated with RNase, dephosphorylated, and rephosphorylated in the presence of ATP [γ-32P]. The resulting material was analyzed by SDS-polyacrylamide gel electrophoresis (SDS-PAGE).

We found that a NELF-E antibody immunoprecipitated the entire NELF complex, as determined by mass spectrometry analysis and Western blotting, in an apparently stoichiometric fashion, allowing us to assess RNA binding by each subunit (*Figure 8A* and *Figure 8—figure supplement 1A–C*, *Figure 8—figure supplement 2*). Bands corresponding to NELF-E and NELF-B/C were readily and reproducibly detected in the radiolabeled sample from both cell lines (*Figure 8B* and *Figure 8—figure supplement 1C*). The intensity of the band for NELF-B/C was less than that observed for the NELF-E band, indicating that NELF-B/C may associate more weakly with RNA than NELF-E. This is consistent with the reported high RNA-binding affinity of the NELF-E RRM domain and our biochemical results (*Pagano et al., 2014*; *Rao et al., 2006*). Immunoprecipitation with a NELF-A antibody produced similar results in Jurkat cells (*Figure 8—figure supplement 1D*). To confirm that NELF-C binds RNA, the NELF-A, -B and -C subunits were cloned into mammalian expression vectors and overexpressed in 293FT cells. Consistent with the native protein, the overexpressed NELF-B and NELF-C subunits bound RNA whereas the NELF-A subunit failed to associate with RNA (*Figure 8C*, *Figure 8—figure supplement 3A–C*). Together these results indicate that NELF-B, -C, and -E all associate with RNA in cells.

## Discussion

Deciphering the mechanism of promoter-proximal Pol II pausing is essential for understanding gene regulation and requires structural information of Pol II elongation complexes bound by DSIF, NELF, and P-TEFb. To this end, structures of the involved multi-protein components are required. Structural information is available for Pol II elongation complexes (*Martinez-Rucobo and Cramer, 2013*), DSIF (*Klein et al., 2011*; *Martinez-Rucobo et al., 2011*), and P-TEFb (*Baumli et al., 2008*; *2012*; *Schulze-Gahmen et al., 2013*; *2014*; *Tahirov et al., 2010*). However, structural information about NELF is lacking, except for the RRM domain of NELF-E (*Rao et al., 2006*; *2008*). To close this gap, we report here the crystal structure of the conserved core NELF subcomplex NELF-AC and the architecture of the 4-subunit, complete NELF complex. We further show that NELF-B and NELF-AC bind single-stranded nucleic acids in vitro, and that NELF-B and NELF-C, in addition to NELF-E, associate with RNA in vivo. These results provide an important step in understanding NELF function and provide the basis for a mechanistic analysis of the role of NELF in promoter-proximal pausing.

From our structural, biochemical and in vivo data, we propose an architectural model for the NELF complex. In the complex, NELF-AC binds to the N-terminal region of NELF-B. The N-terminal region

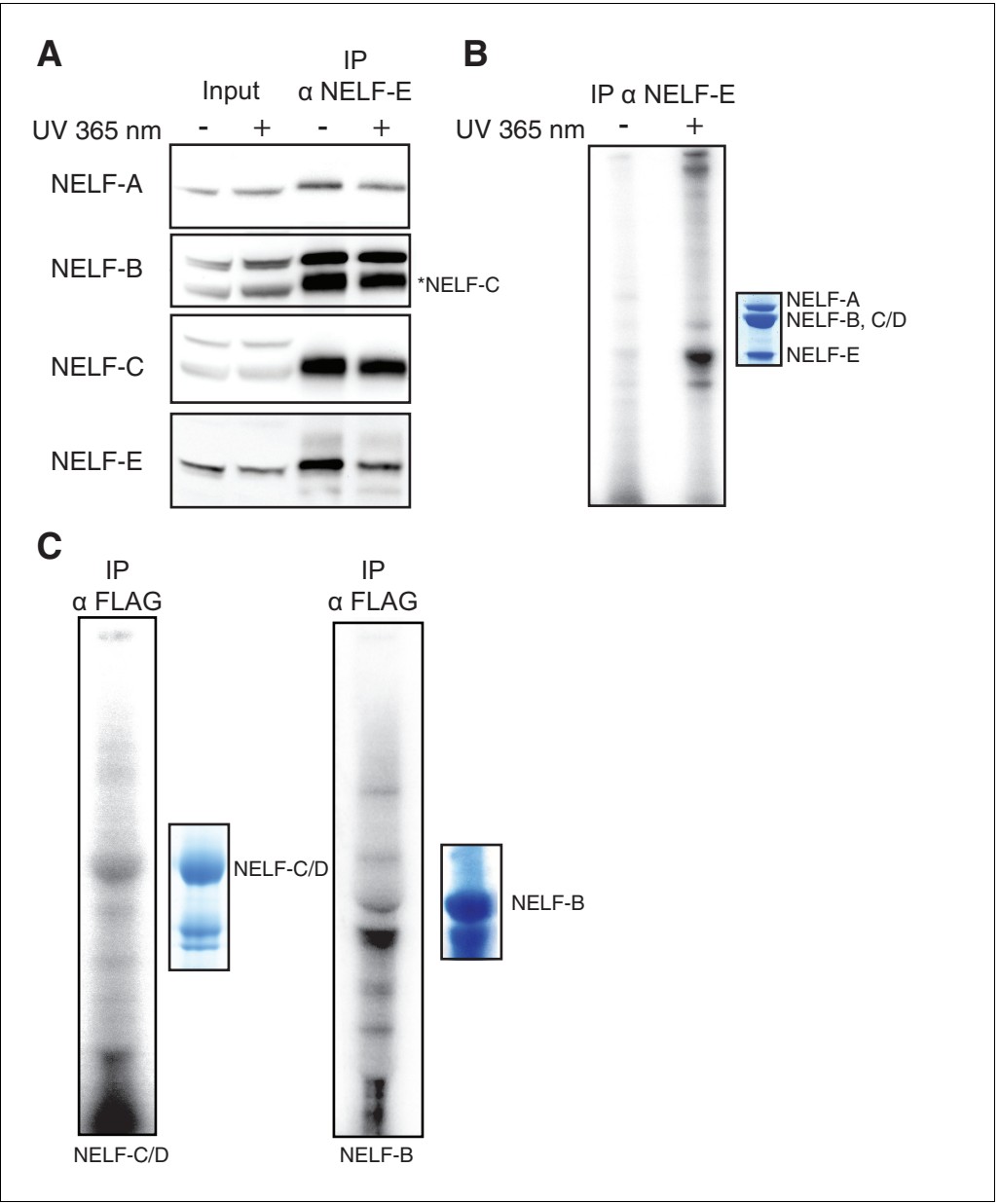

**Figure 8.** NELF association with RNA in cells. (**A**) Immunoprecipitation of the entire NELF complex by a NELF-E specific antibody from Jurkat cells treated with 4-thiouridine (4sU) for 16 hr and UV-crosslinked as detected by Western blot anaylsis with subunit specific antibodies. Cells treated with 4sU, but not UV-crosslinked are shown as a control. Star indicates NELF-C detected on the same blot. Western blot detected with HRP-conjugated secondary antibodies. (**B**) Phosphorimage of SDS-PAGE used to resolve RNAs crosslinked to NELF subunits from Panel A. Protein crosslinked to RNA was treated with PNK in the presence of ATP [γ-32P]. The resulting protein was run on a 4–12% gradient SDS-PAGE. The gel was incubated with a phosphorimage screen and imaged with a Typhoon imager. In crosslinked cells, a band is seen for the NELF-E and NELF-B/C subunits. Cells treated with 4sU, but not UV-crosslinked are shown as a control. A Coomassie stained gel with purified NELF is shown as size reference. (**C**) Phosphorimage of SDS-PAGE used to resolve 5' P-32 labeled RNAs crosslinked to overexpressed -C (left), or -B (right) 293FT cells. Overexpressed NELF-C/-B were immunoprecipitated by N-terminal 3X FLAG tags. Coomassie stained gels of overexpressed NELF-C and -B are shown as a size reference. See ***Figure 8—figure supplement 3*** for further controls.

The following figure supplements are available for figure 8:

**Figure supplement 1.** Additional controls for NELF-B, NELF-C and -E association with RNA in cells.

*Figure 8 continued on next page*

*Figure 8 continued*

**Figure supplement 2.** NELF complex peptides detected by mass spectrometry analysis after NELF-E IP.

**Figure supplement 3.** Overexpression of NELF-A, NELF-B, and NELF-C in 293FT and RNA association.

of NELF-E is sandwiched in between the N- and C- termini of NELF-B, anchoring the flexible NELF-E RRM domain to the rest of the complex. Strong RNA binding by the NELF-E RRM may initially recruit NELF to RNA and secondary binding events by NELF-B and NELF-C may further stabilize the complex on nucleic acid. Future structural studies are required to determine the nucleic acid binding surfaces of NELF-B and to determine how RNA snakes from the NELF-E RRM through the rest of the complex (*Figure 9*). It is also likely that RNA-binding involves major conformational changes.

It is known that the extent of Pol II pausing strongly differs between different genes (*Muse et al., 2007*). Such gene specificity may be explained by differences in promoter-proximal DNA regions. How can DNA sequence influence pausing? First, certain sequences may lead to DNA-RNA hybrids that favor Pol II pausing by slowing down the elongation rate, similar to DNA sequences that influence pausing of bacterial RNA polymerase or *Drosophila* Pol II (*Larson et al., 2014*; *Vvedenskaya et al., 2014*; *Greive and Hippel, 2005*; *Nechaev et al., 2010*). DNA sequence also affects binding of proteins such as GAGA factor and estrogen receptor, which both are reported to recruit NELF to specific genes to induce pausing (*Aiyar et al., 2004*; *Li et al., 2013*). Second, nascent RNA may bind to NELF with different affinities, influencing the efficiency of NELF recruitment to pause sites. Indeed, we observed that nucleic acid binding of NELF-AC may depend on the nucleic acid sequence, and it is known that the RNA-binding activity of the NELF-E RRM domain is sequence-dependent (*Pagano et al., 2014*). It is also known that DNA regions differ in their GC content (*Ginno et al., 2012*) and in *Drosophila* a sequence motif was reported to be associated with pausing (*Hendrix et al., 2008*). Third, nucleosome stabilities vary with DNA sequence and nucleosomes are known to influence Pol II elongation (*Gilchrist et al., 2008*; *2010*; *Mayer et al., 2015*).

NELF association with RNA may not only be required for pausing, but may also be important for mRNA processing when NELF acts together with interaction factors such as Integrator and the cap binding complex (CBC) (*Narita et al., 2007*; *Stadelmayer et al., 2014*; *Yamamoto et al., 2014*). The NELF interaction with CBC is involved in the appropriate 3' processing of histone mRNAs (*Narita et al., 2007*). Similarly, processing of U1, U2, U4, and U5 snRNAs is dependent on NELF and Integrator (*Yamamoto et al., 2014*). Future work is required to determine which RNAs associate with NELF in cells and if this binding activity is independent of NELF's role in promoter proximal pausing.

We note that nucleic acid binding alone may explain recruitment of NELF to certain genes and its association with promoter-proximal regions, but is insufficient to explain Pol II pausing, which additionally requires a change in the elongation behavior of Pol II. This may involve a conformational switch in the polymerase that may be triggered or stabilized by NELF binding to the Pol II surface. Analysis of this intricate mechanism awaits structural studies of functional complexes comprising Pol II, DSIF, NELF and additional factors. The results reported here provide an important step towards this goal.

## Materials and methods

### Preparation of human NELF-AC protein subcomplex for crystallization

The borders of NELF-A and NELF-C within the NELF-AC subcomplex were determined by limited proteolysis of human full-length NELF-AC complex followed by Edman sequencing. Human NELF-A (Q9H3P2) and NELF-C (Q8IXH7) were amplified from codon optimized DNA (Mr. Gene) and cloned into pET28a and pET21b vectors, between *NdeI* and *XhoI* or *NdeI* and *BamHI* restriction sites, respectively, resulting in N-terminally His$_6$-tagged NELF-A (6–188) and untagged NELF-C (183–590). Synthetic oligonucleotides were purchased from Thermo Fisher Scientific and Sigma Genosys.

Plasmids encoding NELF-A (6–188) and NELF-C (183–590) were co-transformed into *E. coli* BL21 CodonPlus (DE3) RIL cells (Stratagene). Cells were grown in LB medium at 37°C until OD600 ~0.6 and cooled on ice for 30 min. Protein expression was induced by the addition of 1 mM IPTG. After

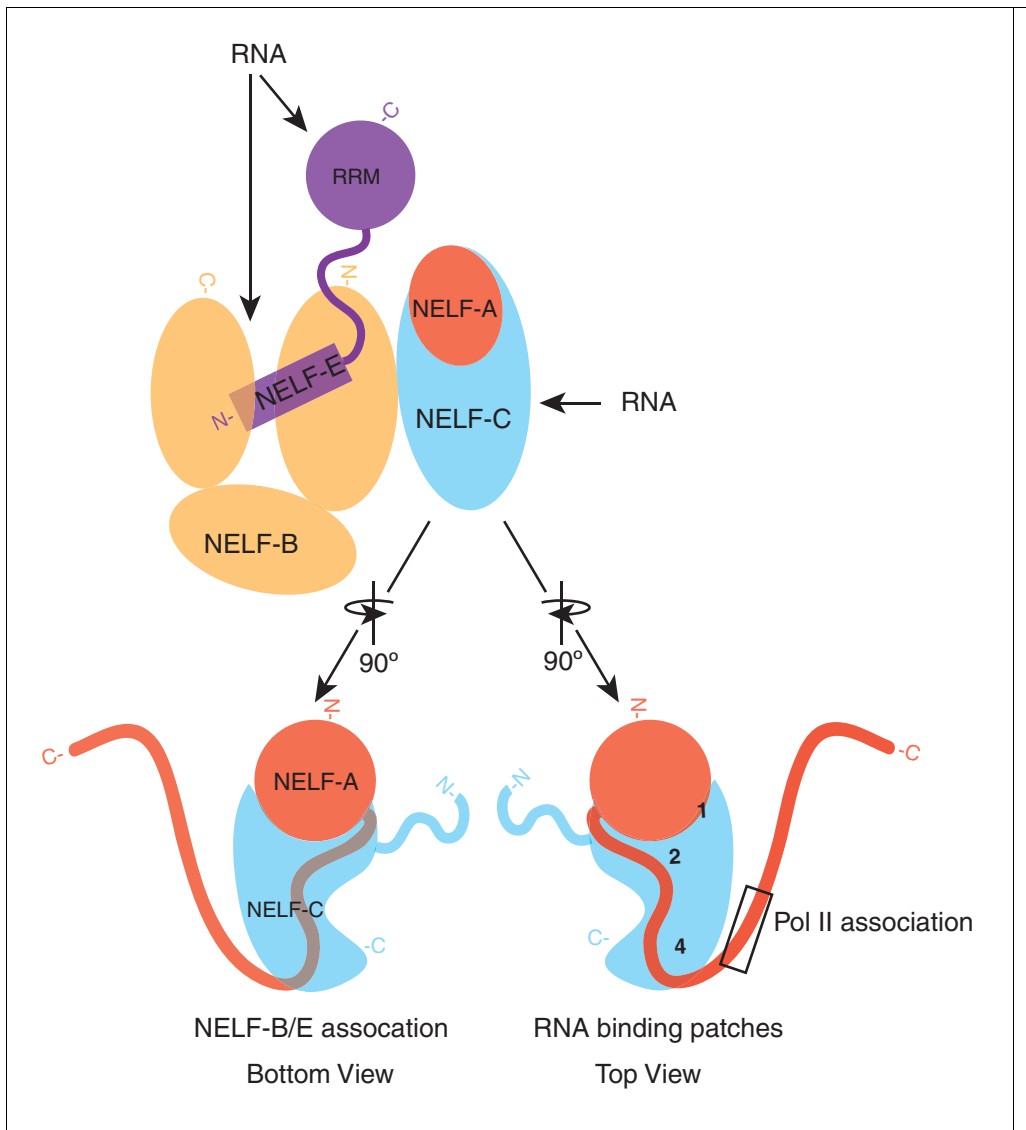

**Figure 9.** Model of NELF architecture and RNA binding regions. The NELF-AC crystal structure, crosslinking data, and in vivo experiments reveal the overall architecture of NELF. NELF-AC (NELF-A red, NELF-C cyan) forms a stable complex that interacts with the N-terminus of NELF-B (orange). NELF-B N- and C- termini sandwich the NELF-E N-terminus. The NELF-E RRM is loosely connected to the entire complex. RNA binds to NELF-E, NELF-B, and NELF-C. The two faces of NELF-AC determined by crystallography and crosslinking are shown below the complex in cartoon format. Three RNA binding faces on the surface of NELF-C are marked with numbers (1, 2, 4). The region of NELF-A that is predicted to bind Pol II, which is absent in our crystal structure is boxed. N- and C-termini for each protein are marked.

induction, cells were grown for an additional 16 hr at 18°C. All purification steps were performed at 4°C. Cells were resuspended and lysed in buffer A (150 mM NaCl, 40 mM Na-HEPES pH 7.4, 10 mM imidazole, 2 mM DTT, 0.284 μg/ml leupeptin, 1.37 μg/ml pepstatin A, 0.17 mg/ml PMSF, 0.33 mg/ml benzamidine). The lysate was applied to Ni-NTA agarose (Qiagen) beads and washed extensively with buffer A containing 20 and 40 mM imidazole. Protein was eluted from the beads with buffer A containing 200 mM imidazole. The eluted protein was mixed with 1 U thrombin/mg protein (Sigma) and dialyzed against buffer B (150 mM NaCl, 40 mM Na-HEPES pH 7.4, 2 mM DTT) for 16 hr at 4°C. The protein was applied to Ni-NTA beads equilibrated in buffer B to remove uncleaved protein. The

Ni-NTA flow through was applied to an anion exchange column (HiTrap Q-HP, 1 ml, GE Healthcare) equilibrated in buffer B. Protein was eluted via a salt gradient from 100 mM to 1 M NaCl in buffer B. The protein was further purified by size exclusion chromatography with the use of a Superose 6 10/300 column (GE Healthcare) equilibrated in buffer B. Peak fractions were pooled and concentrated by centrifugation in Amicon Ultra 4 ml concentrators (10 kDa MWCO) (Millipore) to 12 mg/ml. Protein concentration was determined by absorbance at 280 nm using protein-specific parameters. Protein was aliquoted, flash frozen, and stored at -80°C.

For production of selenomethionine-labeled protein, NELF-AC (6–188 and 183–590) plasmids were co-transformed into *E. coli* B834 (DE3) cells. For protein expression, cells were grown in SelenoMet Medium (AthenES) supplemented with 40 µg/ml L-selenomethionine (SeMet). Selenomethionine-labeled protein was purified as above.

## X-ray structure determination

Native and selenomethionine-labeled NELF-AC crystals were grown by hanging-drop vapor diffusion and were obtained by mixing 1 µl NELF-AC protein (12 mg/ml) with 1 µl reservoir solution containing 14–14.5% (w/v) PEG 3350 and 200 mM sodium malonate pH 6.8–7.0. Tetrahedral NELF-AC crystals grew within 3–5 days. Crystals were cryo-protected in mother liquor containing 25% (w/v) glucose, and flash frozen in liquid nitrogen.

Diffraction data for native crystals were collected under cryo conditions (100 K) in 0.1° increments at beamline X06DA of the Swiss Light Source in Villigen (Switzerland) using a wavelength of 1.0000 Å and a Pilatus 2M-F detector (*Broennimann et al., 2006*). Raw data were processed and scaled with XDS (*Kabsch, 2010*). The structure was solved by SIRAS using diffraction data from an isomorphous crystal of SeMet-labeled protein. Location of 13 selenomethionine sites, calculation of initial phases and density modification were performed with the SHELX suite (*Sheldrick, 2008*). An initial model was built with Buccaneer (*Cowtan, 2006*). The model was iteratively built with COOT (*Emsley and Cowtan, 2004*) and refined with REFMAC (*Vagin et al., 2004*) and phenix.refine (*Adams et al., 2010*) until the R-factors converged. In the final model, 98.2% of residues are in preferred Ramachandran regions and 1.8% of residues are in additionally allowed regions. Figures were prepared with PyMol (*PyMOL, 2002*).

## Fluorescence anisotropy assays with NELF-AC

WT and mutant NELF-AC proteins were expressed and purified as described above. For the final size exclusion step, the column was equilibrated in buffer C (50 mM NaCl, 10 mM HEPES pH 7.4 and 2 mM DTT). Peak fractions were pooled, concentrated by centrifugation to 30 mg/ml, aliquoted, flash frozen, and stored at −80°C.

5′-/6-FAM labeled ssDNA, ssRNA and dsDNA were obtained from Integrated DNA Technologies and dissolved in water to 100 µM. Sequences for ssDNA and dsDNA were 44% GC ACCCCACAAC TAAAAAATCCCAACC, and 60% GC AAGGGGAGCGGGGGAGGATAATAGG (corresponding sequences for ssRNA). Natural ssDNA sequences correspond to sequences of exposed coding (non-template) strands at the *c-fos* gene (bps 87–96 downstream of the TSS [*Fivaz et al., 2000*]) and the *junB* gene (bps 45–54 downstream of the TSS [*Aida et al., 2006*]) during promoter-proximal pausing +/−5 bps (*Figure 5—figure supplement 1C*, inset): AAGACTGAGCCGGCGGCCGC and AGGGAGCTGGGAGCTGGGGG, respectively. Natural ssRNA sequences correspond to 25 nt of nascent mRNA sequence predicted to be proximal to the RNA exit pore on the Pol II surface at *c-fos* (bps 53–77 relative to TSS) and *junB* (bps 13–37 relative to TSS) during promoter-proximal pausing (*Figure 5—figure supplement 1C* inset): CCGCAUCUGCAGCGAGCAUCUGAGA and AGCGGCCAGGCCAGCCUCGGAGCCA, respectively. The sequence corresponding to the HIV-1 TAR RNA stem loop is: CCAGAUCUGAGCCUGGGAGCUCUCUGG. The HIV-1 TAR RNA substrate was diluted to 50 µM with folding buffer (final conditions 100 mM NaCl, 20 mM Na•HEPES pH 7.5, 3 mM MgCl$_2$, 10% (v/v) glycerol). The TAR RNA was folded by incubating the RNA at 95°C for 3 min and transferring to ice for 10 min. The TAR RNA was diluted in folding buffer instead of water for all experiments.

NELF-AC was serially diluted in two fold steps in buffer C. Nucleic acid (2.4 µl, 10 nM final concentration) and NELF-AC (12 µl, 100–0.1 µM final concentration) were mixed on ice and incubated for 10 min. The assay was brought to a final volume of 24 µl and incubated for 20 min at RT in the

dark (final conditions: 30 mM NaCl, 3 mM MgCl$_2$, 10 mM Na-HEPES pH 7.4, 2 mM DTT and 50 µg/ml BSA). To test for non-specific binding, 5 µg/ml yeast tRNA (Sigma) was added to reactions as a competitor (**Figure 5—figure supplement 1**). 20 µl of each solution was transferred to a Greiner 384 Flat Bottom Black Small volume plate.

Fluorescence anisotropy was measured at 30°C with an Infinite M1000Pro reader (Tecan) with an excitation wavelength of 470 nm (±5 nm), an emission wavelength of 518 nm (±20 nm) and a gain of 72. All experiments were done in triplicate and analyzed with GraphPad Prism Version 6. Binding curves were fit with a single site quadratic binding equation:

$$y = \left( \frac{Bmax * \left( [x] + [L] + Kd, app - \sqrt{([x] + [L] + Kd, app)^2 - 4([x] * [L])} \right)}{2 * [L]} \right)$$

where Bmax is the maximum specific binding, L is the concentration of nucleic acid, x is the concentration of NELF-AC, Kd,app is the apparent disassociation constant for NELF-AC and nucleic acid. Error bars are representative of the standard deviation from the mean of three experimental replicates. Experiments were performed on different days from at least two different protein preparations.

## Cloning and expression of human FL NELF tetramer, NELF ΔRRM, NELF-ABC, NELF-B, and NELF-BE (1–138)

Vectors encoding the full-length human cDNAs for NELF-A, -B, -D and -E were generous gift of Hiroshi Handa (**Narita et al., 2003**) and were used as PCR templates for subcloning into modified pFastBac vectors via ligation independent cloning (LIC) (a gift of Scott Gradia, UC Berkeley, vectors 438-A, 438-B [Addgene: 55218, 55219]). NELF-D bears an N-terminal 6x His tag followed by a tobacco etch virus protease cleavage site. Individual subunits were combined into a single plasmid by successive rounds of ligation independent cloning. Each subunit is proceeded by a PolH promoter and followed by an SV40 termination site. For simplicity, residues of the NELF-D subunit are numbered in this text according to NELF-C nomenclature. The NELF-D patch mutant was generated as a synthetic gene block (IDT) from the cDNA sequence and cloned into the 438-B vector using LIC. NELF-E (1–138) was truncated by round the horn PCR. For NELF-B or NELF-BE (1–138) expression constructs, NELF-B was cloned with an N-terminal 6x His tag followed by a tobacco etch virus protease cleavage site.

Purified plasmid DNA (0.5–1 µg) was electroporated into DH10EMBacY cells to generate bacmids (**Berger et al., 2004**). Bacmids were prepared from positive clones by isopropanol precipitation and transfected into Sf9/Sf21 cells grown in Sf-900 III SFM (ThermoFisher) or ESF921 (Expression Technologies), respectively, with X-tremeGENE9 transfection reagent (Sigma) to generate V0 virus. V0 virus was harvested 48–72 hr after transfection. V1 virus was produced by infecting 25 ml of Sf9 or Sf21 cells grown at 27°C, 300 rpm with V0 virus (1E6 cell/ml, 1:50 (v/v) cells:virus). V1 viruses were harvested 48 hr after proliferation arrest and stored at 4°C. For protein expression, 600 ml of Hi5 cells grown in ESF921 medium (Expression Technologies) were infected with 300 µl of V1 virus and grown for 48 hr at 27°C. Cells were harvested by centrifugation (238xg, 4°C, 30 min), resuspended in lysis buffer at 4°C (300 mM NaCl, 20 mM Na•HEPES pH 7.4, 10% glycerol (v/v), 1 mM DTT, 30 mM imidazole pH 8.0, 0.284 µg/ml leupeptin, 1.37 µg/ml pepstatin A, 0.17 mg/ml PMSF, 0.33 mg/ml benzamidine), snap frozen, and stored at −80°C.

## Purification of full-length (FL) NELF tetramer, NELFΔRRM, NELF-ABC, NELF-B, and NELF-BE (1–138)

Protein purification steps were performed at 4°C. Frozen cell pellets were thawed and lysed by sonication. Lysates were clarified by centrifugation in an A27 rotor (ThermoFisher) (26,195 xg, 4°C, 30 min), followed by ultracentrifugation in a Type 45 Ti rotor (Beckman Coulter) (235,000 xg, 4°C, 60 min). Clarified lysates were filtered through 0.8 µm syringe filters (Millipore) and applied to a 5 mlL HisTrap columns (GE Healthcare) equilibrated in lysis buffer. HisTrap columns were washed with 10CV of lysis buffer followed by 5CV of high salt wash buffer (800 mM NaCl, 20 mM Na•HEPES pH 7.4, 10% glycerol (v/v), 1 mM DTT, 30 mM imidazole pH 8.0, 0.284 µg/ml leupeptin, 1.37 µg/ml

pepstatin A, 0.17 mg/ml PMSF, 0.33 mg/ml benzamidine) and 5CV of lysis buffer. The NELF-B construct was washed with a high salt buffer containing 1 M NaCl.

For the FL NELF tetramer, HisTrap columns were washed with 5CV of low salt buffer (150 mM NaCl, 20 mM Na•HEPES pH 7.4, 10% glycerol (v/v), 1 mM DTT, 30 mM imidazole pH 8.0, 0.284 µg/ml leupeptin, 1.37 µg/ml pepstatin A, 0.17 mg/ml PMSF, 0.33 mg/ml benzamidine) before a tandem 5 ml HiTrap Q and HiTrap S column (GE Healthcare) equilibrated in low salt buffer were directly coupled to the HisTrap column. Protein was eluted from the HisTrap column by a gradient from 0–100% nickel elution buffer (150 mM NaCl, 20 mM Na•HEPES pH 7.4, 10% glycerol (v/v), 1 mM DTT, 500 mM imidazole pH 8.0, 0.284 µg/ml leupeptin, 1.37 µg/ml pepstatin A, 0.17 mg/ml PMSF, 0.33 mg/ml benzamidine), after which the HisTrap and HiTrap S column were decoupled from the HiTrap Q column. The HiTrap Q column was washed with 5CV of low salt buffer and protein was eluted by gradient from 0–100% high salt buffer. Peak fractions were analyzed by SDS-PAGE. HiTrap Q fractions containing FL NELF were combined with 2 mg of His$_6$-TEV protease, 416 µg lambda protein phosphatase and dialyzed overnight at 4°C in a Slide-A-Lyzer (2–12 ml 10 kDa MWCO) (Thermo-Fisher) against 1 L of lysis buffer containing 1 mM MnCl$_2$. Truncation constructs were eluted directly from the HisTrap column by a gradient from 0–100% nickel elution buffer containing 300 mM NaCl. Peak fractions were analyzed by SDS-PAGE, and pooled for TEV protease and lambda phosphatase treatment overnight as described for the FL tetrameric protein.

Protein was removed from the Slide-A-Lyzer cassette and applied to a 5 mL HisTrap column to remove uncleaved protein and TEV protease. Protein was concentrated in an Amicon 15 ml centrifugal concentrator (FL tetramer 100 MWCO; NELF ΔRRM and NELF-ABC 50 MWCO; NELF-B and NELF-BE (1–138) 30 MWCO) (Millipore) to 1.0–2.0 ml. The protein was applied to a S200 16/600 pg column (GE Healthcare) equilibrated in 150 mM NaCl, 20 mM Na•HEPES pH 7.4, 10% (v/v) glycerol, and 1 mM DTT. Peak fractions were analyzed by SDS-PAGE. Pure fractions were concentrated as described above to 500 µl, aliquoted, flash frozen, and stored at −80°C. Typical protein preparations yield 10–15 mg of FL tetrameric NELF from 1 L of insect cell culture.

## Fluorescence anisotropy of FL NELF tetramer, NELFΔRRM, NELF-ABC, NELF-B, and NELF-BE (1–138)

Fluorescence anisotropy experiments were performed essentially as described for NELF-AC except for the following modifications. Protein was diluted in half log dilution steps in a buffer containing 150 mM NaCl, 20 mM Na•HEPES pH 7.4, 10% (v/v) glycerol, and 1 mM DTT. The final buffer contained 60 mM NaCl, 3 mM MgCl$_2$, 10 mM Na-HEPES pH 7.4, 2 mM DTT, 50 µg/ml BSA, and 5 µg/ml baker's yeast tRNA. 18 µl of the solution was used for measurements.

## Mass spectrometric identification of crosslinking sites

The 4-subunit human NELF complex (10 µg, 5.5 µM in 95 µl final volume) purified from insect cells was incubated with 1.1 mM disuccinimidyl suberate (DSS) H12/D12 (Creative Molecules) for 30 min at 30°C in a final buffer containing 100 mM NaCl, 30 mM Na•HEPES pH 7.4, 10% (v/v) glycerol, 1 mM DTT, and 3 mM MgCl$_2$. The crosslinking reaction was quenched by adding ammonium bicarbonate to a final concentration of 100 mM and incubation for 10 min at 30°C. The chemical cross-links on NELF complexes were identified by mass spectrometry as described previously (*Herzog et al., 2012*). Briefly, cross-linked complexes were reduced with 5 mM TCEP (Thermo Scientific) at 35°C for 15 min and subsequently treated with 10 mM iodoacetamide (Sigma-Aldrich) for 30 min at room temperature in the dark. Digestion with lysyl enodpeptidase (Wako) was performed at 35°C, 6 M Urea for 2 hr (at enzyme-substrate ratio of 1:50 w/w) and was followed by a second digestion with trypsin (Promega) at 35°C overnight (also at 1:50 ratio w/w). Digestion was stopped by the addition of 1% (v/v) trifluoroacetic acid (TFA). Acidified peptides were purified using C18 columns (Sep-Pak, Waters). The eluate was dried by vacuum centrifugation and reconstituted in water/acetonitrile/TFA, 75:25:0.1. Cross-linked peptides were enriched on a Superdex Peptide PC 3.2/30 column (300 × 3.2 mm) at a flow rate of 25 µl min−1 and water/acetonitrile/TFA, 75:25:0.1 as a mobile phase. Fractions of 100 µl were collected, dried, and reconstituted in 2% acetonitrile and 0.2% FA, and further analyzed by liquid chromatography coupled to tandem mass spectrometry using a hybrid LTQ Orbitrap Elite (Thermo Scientific) instrument. Cross-linked peptides were identified using xQuest (*Walzthoeni et al., 2012*). False discovery rates (FDRs) were estimated by using

xProphet (*Walzthoeni et al., 2012*) and results were filtered according to the following parameters: FDR = 0.05, min delta score = 0.90, MS1 tolerance window of −4 to 4 ppm, ld-score > 22.

## Analysis of NELF-RNA binding in cells

Jurkat cells were maintained in RPMI (Gibco) with 10% FBS and Glutamax. 293FT cells were maintained in DMEM (Gibco) with 10% FBS and Glutamax. Cells were routinely checked for mycoplasma contamination using the PlasmoTest Mycoplasma Detection Kit (InvivoGen, 12K06-MM). Antibodies used were anti-NELF-A (Santa Cruz, sc-23599); anti-COBRA1 (Bethyl Laboratories, A301-911A-M); anti-THL1 (Cell Signaling, D5G6W); anti-NELF-E (Millipore, ABE48); anti-GAPDH (Sigma, G8795); anti-FLAG M2 (Sigma, clone M2, F1804); and anti-c-MYC (Sigma, clone 9E10, M4439).

Photoactivatable-ribonucleoside-enhanced crosslinking and immunoprecipitation was performed as described, with a few modifications. All concentrations are final unless otherwise indicated. Jurkat and 293FT cells were incubated with 4-thiouridine (4sU) (100 µM) (Carbosynth, NT06186) for 16 hr in growth medium. Cells were then washed two times with PBS and crosslinked at 365 nm with a Bio-Link BLX 365 (PeqLab) UV lamp operated at 0.15 J/cm$^2$ (293FT) or 0.2 J/cm$^2$ (Jurkat). Cells were scraped from the plates with cold PBS and collected by centrifugation. Pellets were resuspended in 3 volumes of NP-40 Lysis Buffer (50 mM HEPES-KOH pH 7.5, 150 mM KCl, 2 mM EDTA-NaOH pH 8.0, 1 mM NaF, 0.5% (v/v) NP-40, 0.5 mM DTT, 1x complete EDTA-free protease inhibitor cocktail (Sigma, P8340)) and incubated on ice for 10 min. The resulting lysate was then passed through a syringe with 27G needle seven times and centrifuged at 13000 *g* for 15 min at 4°C. The supernatant fraction was further clarified by passing it through a 5 µm syringe filter (Pall Corporation, 4650). Total protein concentration was determined by the Bradford method. Control cells were treated with 4sU but not crosslinked and underwent the same treatment as the crosslinked samples.

For endogenous NELF-A and NELF-E immunoprecipitation experiments, 30 mg of total protein was incubated with 70 µg of anti-NELF-A antibody (Santa Cruz, sc-23599) or 50 µg of anti-NELF-E antibody (Bethyl Laboratories, A301-914A) conjugated to Protein G Dynabeads (Invitrogen, B00262). For plasmid overexpression experiments, 20 mg of total protein was incubated with 70 µl anti-FLAG M2 Magnetic Beads (Sigma, M8823) or 15 µg of anti-c-Myc antibody (Sigma, clone 9E10, M4439) conjugated to Protein G Dynabeads (Invitrogen, B00262). The protein was incubated with the antibody-conjugated beads for 2 hr at 4°C on a rotating wheel. Beads were then washed three times with 1 ml of cold IP Wash Buffer (50 mM HEPES-KOH pH 7.5, 300 mM KCl, 0.05% (v/v) NP-40, 0.5 mM DTT, 1x complete EDTA-free protease inhibitor cocktail (Sigma, P8340)). The beads were resuspended in 200 µl of IP Wash Buffer and treated with 50 U/µl RNAse T1 (Thermo Scientific, EN0542) for 15 min at 22°C and cooled on ice for 5 min. Beads were washed three times with 1 ml of cold High Salt Wash Buffer (50 mM HEPES-KOH pH 7.5, 500 mM KCl, 0.05% (v/v) NP-40, 0.5 mM DTT, 1x complete EDTA-free protease inhibitor cocktail (Sigma, P8340)), followed by one wash with 1 ml of Phosphatase Buffer pH 6.0 (50 mM Tris-HCl pH 7.0, 1 mM Mg$_2$Cl$_2$, 0.1 mM ZnCl$_2$).

RNAs were dephosphorylated in 100 µl of Phosphatase Reaction Mix (1X Antarctic Phosphatase Reaction Buffer (NEB, M0289S), 1 U/µl Antarctic Phosphatase (NEB, M0289S), and 1 U/µl RNase OUT (Invitrogen, 10777–019)) for 30 min at 37°C, 300 rpm. Beads were washed once with 1 ml of Phosphatase Wash Buffer (50 mM Tris-HCl pH 7.5, 20 mM EGTA, 0.5% (v/v) NP-40) and two times in Polynucleotide Kinase Buffer (50 mM Tris-HCl pH 7.5, 50 mM NaCl, 10 mM MgCl$_2$). Beads were resuspended in 20 µl of Kinase Reaction Mix (1X T4 PNK Reaction Buffer (NEB, M0201S), 1 U/µl T4 PNK (NEB, M0201S), 2 U/µl RNAse OUT (Invitrogen, 10777–019), and 1 µCi/µl ATP-γ-32P (Perkin Elmer, NEG502Z) and incubated for 1 hr at 37°C, 800 rpm. Beads were washed five times with 1 ml of Polynucleotide Kinase Buffer (50 mM Tris-HCl pH 7.5, 50 mM NaCl, 10 mM MgCl$_2$), resuspended in 25 µl of 2X SDS-loading buffer, and incubated for 5 min at 95°C. The eluted supernatant (20 µl) was run on a Novex Bis-Tris 4–12% (Invitrogen) SDS-PAGE in 1X MOPS buffer for 1 hr at 160 V. The gel was exposed to a phosphorimager screen overnight at −20°C. The phosphorimager screen was scanned on a Typhoon FLA 9500 (GE). To determine the specificity and crossreactivity of the NELF antibodies used for immunoprecipitation experiments, samples treated with nonradioactive ATP were submitted for mass spectrometry (MS) analysis. The MS analysis confirmed that all NELF subunits are present and verified that the detected radiolabeled signal corresponds to NELF subunits.

To generate NELF-A, -B, and -C overexpression plasmids, the gene coding regions of human NELF-A, -B, and -C were cloned into pCMV-GLuc2 (NEB) between the BamHI and NotI sites. Genes were cloned with N-terminal affinity tags followed by a TEV protease cleavage site (3xMYC NELF-A,

3xFLAG NELF-B and -C). DNA for transfection was isolated from *E. coli* and purified by phenol chloroform extraction and ethanol precipitation. Plasmids were resuspended in water at a final concentration of ~10 µg/µL and stored at −20°C prior to transfection.

For plasmid transfection and 4-thiouridine labeling in 293FT, plasmids were transfected into 293FT cells using Lipofectamine 2000 (ThermoFisher Scientific) as directed by the manufacturer. Briefly, 25 µg of plasmid was transfected into cells growing in p145 $cm^2$ dishes (5 dishes per condition). Thirty-two hrs after transfection, 100 µM 4sU (Carbosynth, NT06186) was added to the growth medium. Cells were incubated at 37°C for an additional 14–16 hr. Photoactivatable-ribonucleoside-enhanced crosslinking and immunoprecipitation experiments were performed 48 hr after transfection.

## Acknowledgements

We thank the crystallization facility at the Max Planck Institute of Biochemistry, Martinsried, Germany. Part of this work was performed at the Swiss Light Source at the Paul Scherrer Institut, Villigen, Switzerland. We thank Monika Raabe, Annika Kühn and Henning Urlaub for help with mass spectrometry. We thank the Cramer lab for discussions. We thank Oleh Rymarenko for supporting initial cloning of NELF truncation mutants. SMV is supported by an EMBO Long-Term Postdoctoral Fellowship (ALTF 745-2014). LC is supported by an EMBO Long-Term Postdoctoral Fellowship (ALTF 1261-2014). FH received funding from the European Research Council (StG no. 638218) and the Deutsche Forschungsgemeinschaft (GRK1721). PC was supported by the Deutsche Forschungsgemeinschaft (SFB 860), the European Research Council Advanced Grant TRANSIT, and the Volkswagen Foundation.

## Additional information

### Funding

| Funder | Grant reference number | Author |
| --- | --- | --- |
| European Molecular Biology Organization | ALTF 745-2014 | Seychelle M Vos |
| European Molecular Biology Organization | ALTF 1261-2014 | Livia Caizzi |
| Deutsche Forschungsgemeinschaft | GRK1721 | Franz Herzog |
| European Research Council | StG 638218 | Franz Herzog |
| European Research Council | TRANSIT | Patrick Cramer |
| Deutsche Forschungsgemeinschaft | SFB860 | Patrick Cramer |
| Volkswagen Foundation | | Patrick Cramer |

The funders had no role in study design, data collection and interpretation, or the decision to submit the work for publication.

### Author contributions

SMV, Conception and design, Acquisition of data, Analysis and interpretation of data, Drafting or revising the article; DP, LC, KBH, Acquisition of data, Analysis and interpretation of data, Drafting or revising the article; PR, TZ, FH, Acquisition of data, Analysis and interpretation of data; PC, Conception and design, Analysis and interpretation of data, Drafting or revising the article

### Author ORCIDs

Patrick Cramer, http://orcid.org/0000-0001-5454-7755

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
