## [Decision Letter]

Thank you for submitting your article "NELF architecture and RNA binding" for consideration by *eLife*. Your article has been reviewed by three peer reviewers, one of whom is a member of our Board of Reviewing Editors and the evaluation has been overseen by Kevin Struhl as the Senior Editor.

The reviewers have discussed the reviews with one another and the Reviewing Editor has drafted this decision to help you prepare a revised submission.

Note: The authors should make their title more broadly accessible.

– Use of acronyms (like NELF) is discouraged. NELF should be described as the Negative Elongation Factor

– The authors should note that they are working with human NELF proteins

(further information is below)

Summary:

NELF complex plays a critical role in RNA polymerase II pausing, which has been shown to be important in transcriptional regulation of many genes in various organisms. The structure of NELF complex, which has been elusive for a long time, is crucial of mechanistic understanding of its function in Pol II pausing. In their paper, Patrick Cramer and his colleagues have solved the crystal structure of NELF-A/C subcomplex, and modeled NELF-B and -E based on the cross-linking data. Interestingly, in addition to the predominant RNA interaction of NELF complex via the RRM domain of NELF-E subunit, they identified a region of NELF-A/C complex that associates with RNA (and ssDNA) both in vitro and in vivo. Information provided in this manuscript is a first step for a detailed mechanistic study of NELF's role in Pol II pausing and its regulation.

While the study provides novel insights into the assembly structure of the NELF complex, there are several limitations (listed below), that should be addressed prior to publication.

Additionally, the reviewers noted that there is limited explanation of figures in the main text, and strongly recommend that the authors endeavor to describe the experiments and figures in more detail so that a reader needn't access supplemental methods in order to understand what has been done.

Essential revisions:

1) The authors identified novel RNA binding sites in NELF-C, but it is unclear whether this interaction actually occurs in the context of tetramer (the authors observed RNA-crosslinking to NELF-C only when over-expressing NELF-C, Figure 6—figure supplement 1) and whether this interaction is important for the NELF function. Crosslinking data obtained with full NELF complex purified from insect cells (data not shown) is arguably the most informative about the structure of NELF complex, compared to data obtained from other incomplete NELF subcomplexes. It would be important to include this data in this manuscript, ideally as Figure 5.

2) It was unclear to the reviewers whether the authors were arguing for sequence specificity in the RNA binding of NELF-C (Figure 4). However, we felt that the data did not support specificity. The affinity of NELF-AC binding to 'random' single stranded nucleic acids in Figure 4 is quite low, but it is just as good as the presumed 'specific' targets shown in Figure 4.

Since this RNA binding region of NELF-C is a main novel aspect of the manuscript, the authors should clarify this section. If they believe there is any specificity to the RNA ligands bound, they must expand their biochemical analysis of NELF-AC binding to nucleic acids. For example, Figure 4 should have several negative controls, such as sequences that occur in *junB* or *fos* well downstream of the pause site, sequences that won't form simple single stranded regions, such as hairpins, etc. However, if the authors concur that the RNA binding is likely non-specific charge-charge interactions, this should be clearly spelled out in the manuscript.

For example, in subsection “NELF-AC binds single-stranded nucleic acids” "both the strength of binding and the preference for RNA or DNA are sequence-dependent." In this concluding sentence of the paragraph, the authors should add "protein" before "sequence" or substitute "protein residue-dependent" to make it clear that they are referring to the protein having regions that are specific for RNA binding and not that the protein binds specific RNAs.

3) The data supporting an interaction of NELF-C with RNA in cells is not very strong. First, the bands for NELF-C/D are not well separated from NELF-B, making it totally possible that they are in fact detecting NELF-B interactions with RNA rather than NELF-C. This is a serious caveat. Further, I am concerned that the signals shown don't appear to depend much on UV irradiation (especially in 293FT cells, Figure 6—figure supplement 1). Can the authors find a more convincing way to demonstrate NELF-C interactions with RNA in vivo? This would really strengthen the manuscript.

4) The authors should provide a final model of the NELF architecture with and without RNA. The authors may also consider performing protein-protein cross linking (as in Figure 5) in the presence of RNA to understand the architecture of the tetramer bound to RNA, and to better understand the protein-RNA cross linking result.

[Editors' note: further revisions were requested prior to acceptance, as described below.]

Thank you for resubmitting your work entitled "Architecture and RNA binding of the human Negative Elongation Factor" for further consideration at *eLife*. Your revised article has been favorably evaluated by Kevin Struhl (Senior editor) and a Reviewing editor. The manuscript has been improved but there are some remaining issues that need to be addressed before acceptance, as outlined below:

In the revised version of Vos et al. the authors do a nice job of clarifying the manuscript and addressing reviewer concerns. Overall, the manuscript is now acceptable for publication. The presentation of a basic structural model for NELF is highly useful for the field, as is the discovery of multiple, low-affinity RNA binding sites across the surface of the NELF complex. This work will clearly provide a nice framework for extending our mechanistic understanding of NELF activity and will stimulate considerable follow-up.

However, there is one remaining issue we would like the authors to remedy: the language about the specificity of RNA binding by NELF-AC is stronger than warranted by the data presented. NELF-AC does appear to bind RNA comprised of 60% GC more strongly than 40% GC or TAR RNA, but the nature of such selectivity is not addressed (What might be driving this preference? Is it a question of structure? Flexibility? A specific motif? These questions are all left unanswered). Thus, there is concern over definitive statements such as "the strength of binding and preference for RNA or DNA are sequence-dependent".

When one talks of specificity in an RNA-binding protein, one often is referring to a specific motif or sequence context (see work on PUF proteins, or NELF-E, etc.) that is recognized by specific interactions. That is not at all the case in this work. For this reason, we request that the authors modify their language in subsection “NELF-AC binds single-stranded nucleic acids” and elsewhere to better match the limited data presented. Rather than their data 'indicating' specificity, rather it 'suggests' specificity, or they 'provide some evidence for selectivity'. If the authors can tone down this language to better reflect the data presented, we will be happy to accept this work.

---

## [Author Response]

The reviewers have discussed the reviews with one another and the Reviewing Editor has drafted this decision to help you prepare a revised submission.

Note: The authors should make their title more broadly accessible.

- Use of acronyms (like NELF) is discouraged. NELF should be described as the Negative Elongation Factor

*- The authors should note that they are working with human NELF proteins*

(further information is below)

We have altered the title of the manuscript to make it more accessible to the community. The Abstract, Introduction, and Materials and methods section have been modified to make it clear that the work described in this manuscript was conducted with human NELF proteins.

Summary:

NELF complex plays a critical role in RNA polymerase II pausing, which has been shown to be important in transcriptional regulation of many genes in various organisms. The structure of NELF complex, which has been elusive for a long time, is crucial of mechanistic understanding of its function in Pol II pausing. In their paper, Patrick Cramer and his colleagues have solved the crystal structure of NELF-A/C subcomplex, and modeled NELF-B and -E based on the cross-linking data. Interestingly, in addition to the predominant RNA interaction of NELF complex via the RRM domain of NELF-E subunit, they identified a region of NELF-A/C complex that associates with RNA (and ssDNA) both in vitro and in vivo. Information provided in this manuscript is a first step for a detailed mechanistic study of NELF's role in Pol II pausing and its regulation.

While the study provides novel insights into the assembly structure of the NELF complex, there are several limitations (listed below), that should be addressed prior to publication.

Additionally, the reviewers noted that there is limited explanation of figures in the main text, and strongly recommend that the authors endeavor to describe the experiments and figures in more detail so that a reader needn't access supplemental methods in order to understand what has been done.

We thank the referees for their insightful comments and suggestions. The suggestions have led to a much improved manuscript. We present several new experiments in the manuscript including crosslinking-mass spectrometric analysis with the full-length (FL) NELF tetramer produced in insect cells, RNA binding experiments with the FL NELF tetramer, and in vivo experiments after NELF-B overexpression. Our conclusions could be expanded with the new data. Additional information concerning the figures has been added to the text and figure legends to make things clearer for the reader. We have addressed the points raised by the reviewers.

Essential revisions:

1) The authors identified novel RNA binding sites in NELF-C, but it is unclear whether this interaction actually occurs in the context of tetramer (the authors observed RNA-crosslinking to NELF-C only when over-expressing NELF-C, Figure 6—figure supplement 1) and whether this interaction is important for the NELF function. Crosslinking data obtained with full NELF complex purified from insect cells (data not shown) is arguably the most informative about the structure of NELF complex, compared to data obtained from other incomplete NELF subcomplexes. It would be important to include this data in this manuscript, ideally as Figure 5.

We have performed a number of additional experiments to address this important point. Fluorescence anisotropy experiments were performed with the complete NELF complex and subcomplexes of NELF containing NELF-ABC or NELF-ABC and an -E variant lacking the RRM. These experiments reveal that NELF binds to RNA using regions outside of the well-characterized NELF-E RRM. Introduction of NELF-C patch mutations in the tetramer reduce RNA association of the complex, further supporting NELF-C dependent RNA interactions within the tetramer. We also find that NELF-B binds to RNA with a similar affinity as measured for NELF-AC. As advised by the referees, we performed additional crosslinking-MS experiments with the complete NELF complex purified from insect cells. This data is now reported in the manuscript and indeed indicates that in the context of the tetramer, the reported RNA binding face of NELF-C is not involved in contacts between other subunits.

2) It was unclear to the reviewers whether the authors were arguing for sequence specificity in the RNA binding of NELF-C (Figure 4). However, we felt that the data did not support specificity. The affinity of NELF-AC binding to 'random' single stranded nucleic acids in Figure 4 is quite low, but it is just as good as the presumed 'specific' targets shown in Figure 4.

Since this RNA binding region of NELF-C is a main novel aspect of the manuscript, the authors should clarify this section. If they believe there is any specificity to the RNA ligands bound, they must expand their biochemical analysis of NELF-AC binding to nucleic acids. For example, Figure 4 should have several negative controls, such as sequences that occur in junB or fos well downstream of the pause site, sequences that won't form simple single stranded regions, such as hairpins, etc. However, if the authors concur that the RNA binding is likely non-specific charge-charge interactions, this should be clearly spelled out in the manuscript.

For example, in subsection “NELF-AC binds single-stranded nucleic acids” "both the strength of binding and the preference for RNA or DNA are sequence-dependent." In this concluding sentence of the paragraph, the authors should add "protein" before "sequence" or substitute "protein residue-dependent" to make it clear that they are referring to the protein having regions that are specific for RNA binding and not that the protein binds specific RNAs.

We apologize for the lack of clarity in this section. Indeed, we observe that NELF-AC prefers to bind certain RNA sequences. We have added evidence to confirm this point. Use of another random 25 nt RNA/DNA sequence resulted in no detectable binding by the NELF-AC complex. This data has now been incorporated as panel 5B. Further, the HIV-1 TAR RNA was previously shown to be a preferred substrate for the *Drosophila* and human NELF-E RRM (Pagano et al., 2014, Yamaguchi et al., 2002, Fuijnaga et al., 2004) in vitro. To expand upon these findings, we tested whether the stem loop of the HIV-1 TAR RNA associates with NELF through regions outside of the RRM. Indeed, we find that NELF associates with the TAR stem loop in the absence of NELF-E. From our data, it appears that NELF-B in complex with a fragment of -E modestly associates with the RNA. In contrast, NELF-AC does not associate with the TAR RNA, further suggesting that its ability to associate with RNA is sequence-dependent.

We made sure the conclusions reflect correctly what is observed. We had performed binding experiments with substrates that correspond to pausing sequences of *JunB* and *c-Fos* and determined binding constants that are similar to those obtained with the 60% GC random sequence. That we observe interactions with these sequences in vitro does not mean that NELF-AC associates with these sequences in vivo. We only interpret these data to mean that NELF-AC has a propensity to associate with sequences associated with the *JunB* or *c-Fos*. We have moved these figures to the supplementary material.

*3) The data supporting an interaction of NELF-C with RNA in cells is not very strong. First, the bands for NELF-C/D are not well separated from NELF-B, making it totally possible that they are in fact detecting NELF-B interactions with RNA rather than NELF-C. This is a serious caveat. Further, I am concerned that the signals shown don't appear to depend much on UV irradiation (especially in 293FT cells, Figure 6—figure supplement 1). Can the authors find a more convincing way to demonstrate NELF-C interactions with RNA* in vivo*? This would really strengthen the manuscript*.

We thank the reviewers for spotting this and share their concerns. NELF-B and -C are of similar MW (66kDa versus 65kDa) and are thus difficult to separate by SDS-PAGE. Unfortunately, we are limited in the number of available technologies for detecting protein•RNA interactions in vivo. PAR-CLIP is well suited but is limited by the availablility of good antibodies to immunoprecipitate (IP) specific complexes or proteins. We attempted to IP NELF-C specifically, but the IP failed with several different antibodies. We also tried to IP NELF-B specifically, but this also failed. We performed PAR-CLIP in several cell lines with the NELF-E IP (HeLa, K562, Jurkat, 293FT). In K562 and HeLa cells, low abundance of the NELF complex and other RNA binding contaminants, respectively, stymied our attempts to orthogonally test whether we could detect a signal for NELF-B/C.

Therefore, to determine whether NELF-B could contribute to the radiolabeled signal, we performed in vitro binding experiments with NELF-B and found that it indeed associates with RNA (see point 1). This indicates that in our IP from Jurkat cells, the signal we detect can be attributed to either NELF-B or NELF-C or both. We have noted this in the text.

We also performed overexpression experiments in 293FT cells with NELF-B and detect a signal for NELF-B. Further, we have observed that 293FT cells show less signal after UV exposure than Jurkat cells. UV irradiation enhances the RNA crosslinking to proteins but the efficiency of crosslinking might differ for each cell line. It is possible that Jurkat cells possess more RNA bound NELF than 293FT, resulting in a more intense signal after UV irradiation. Taken together, we hope the reviewers appreciate the technical limitations of investigating RNA binding to NELF subunits in vivo, and agree that we show convincingly that NELF-B and -C bind RNA in cells, but certainly there is in vivo RNA binding outside of NELF-E.

4) The authors should provide a final model of the NELF architecture with and without RNA. The authors may also consider performing protein-protein cross linking (as in Figure 5) in the presence of RNA to understand the architecture of the tetramer bound to RNA, and to better understand the protein-RNA cross linking result.

We have provided a model of NELF architecture with and without RNA as Figure 9. We performed protein•protein cross linking of NELF in the presence of a TAR RNA substrate as suggested by the referees. From these data, we detect fewer protein-protein crosslinks, indicating that the RNA reduces the flexibility of some protein regions. We also observe changes in the intraprotein crosslinks when RNA is present, indicating conformational changes in NELF upon RNA binding. We currently lack an atomic model of NELF-B and the N-terminus of NELF-E, which makes it difficult to know exactly how NELF interacts with RNA. We have thus decided to not present the new crosslinking MS data in the current manuscript. We have added more to the Discussion section to describe potential roles for RNA binding by the NELF complex.

[Editors' note: further revisions were requested prior to acceptance, as described below.]

In the revised version of Vos et al. the authors do a nice job of clarifying the manuscript and addressing reviewer concerns. Overall, the manuscript is now acceptable for publication. The presentation of a basic structural model for NELF is highly useful for the field, as is the discovery of multiple, low-affinity RNA binding sites across the surface of the NELF complex. This work will clearly provide a nice framework for extending our mechanistic understanding of NELF activity and will stimulate considerable follow-up.

However, there is one remaining issue we would like the authors to remedy: the language about the specificity of RNA binding by NELF-AC is stronger than warranted by the data presented. NELF-AC does appear to bind RNA comprised of 60% GC more strongly than 40% GC or TAR RNA, but the nature of such selectivity is not addressed (What might be driving this preference? Is it a question of structure? Flexibility? A specific motif? These questions are all left unanswered). Thus, there is concern over definitive statements such as "the strength of binding and preference for RNA or DNA are sequence-dependent".

When one talks of specificity in an RNA-binding protein, one often is referring to a specific motif or sequence context (see work on PUF proteins, or NELF-E, etc.) that is recognized by specific interactions. That is not at all the case in this work. For this reason, we request that the authors modify their language in subsection “NELF-AC binds single-stranded nucleic acids” and elsewhere to better match the limited data presented. Rather than their data 'indicating' specificity, rather it 'suggests' specificity, or they 'provide some evidence for selectivity'. If the authors can tone down this language to better reflect the data presented, we will be happy to accept this work.

We thank you for your positive and kind remarks. We agree that to define the specificity of RNA binding by NELF-AC further experiments must be conducted. We have thus adapted the language in the manuscript to be less definitive and better reflect the data presently available.